# Activating hidden signals by mimicking cryptic sites in a synthetic extracellular matrix

Yumeng Zhu [1,7], Yulia Shmidov [2,6,7], Elizabeth A. Harris [3], Michelle H. Theus [3,4], Ronit Bitton [2,5] ✉ & John B. Matson [1,4] ✉

Cryptic sites are short signaling peptides buried within the native extracellular matrix (ECM). Enzymatic cleavage of an ECM protein reveals these hidden peptide sequences, which interact with surface receptors to control cell behavior. Materials that mimic this dynamic interplay between cells and their surroundings via cryptic sites could enable application of this endogenous signaling phenomenon in synthetic ECM hydrogels. We demonstrate that depsipeptides ("switch peptides") can undergo enzyme-triggered changes in their primary sequence, with proof-of-principle studies showing how trypsin-triggered primary sequence rearrangement forms the bioadhesive pentapeptide YIGSR. We then engineered cryptic site-mimetic synthetic ECM hydrogels that experienced a cell-initiated gain of bioactivity. Responding to the endothelial cell surface enzyme aminopeptidase N, the inert matrix transformed into an adhesive synthetic ECM capable of supporting endothelial cell growth. This modular system enables dynamic reciprocity in synthetic ECMs, reproducing the natural symbiosis between cells and their matrix through inclusion of tunable hidden signals.

The ECM, a network of proteins and polysaccharides that structurally supports cells and organs[1], contains buried peptidic signals called cryptic sites[2], which are inaccessible to cells until enzymatic degradation[3,4] or mechanical stress[5] triggers a structural alteration to uncover these short peptides[5–8]. Once revealed, cryptic site peptide sequences become available to bind specific cellular receptors, providing instructions to cells to initiate behavioral changes, making them a vital component in the cell–ECM synergy[2,9–12]. For example, proteolytic cleavage of collagen IV by MMP-9 exposes a cryptic site hidden within its triple helical structure that is required for angiogenesis[3]. Synthetic hydrogels offer the possibility of mimicking the structural features of native ECM, with vast potential biomedical applications[13–17]. These constructs range from static hydrogels to dynamic

biofunctional systems designed to direct cell activity[18–22]. However, no synthetic hydrogels currently exist that recreate cell-driven changes to matrix bioactivity arising from revelation of natural cryptic sites.

We envisioned that cryptic sites could be mimicked in synthetic hydrogels through the use of so-called "switch peptides,"[23–25] depsipeptides that contain a Ser or Thr residue linked to the peptide backbone through its side chain alcohol via an ester bond. A protecting group is installed on the pendant α-amine, which upon removal leads to an oxygen to nitrogen (O → N) acyl shift, turning the non-functional primary sequence of the depsipeptide into a functional, amide-linked peptide[26]. We set out to synthesize a switch peptide that would rearrange into the bioadhesive YIGSR pentapeptide upon enzymatic activation (deprotection) by addition of trypsin and could be easily

[1]Department of Chemistry and Macromolecules Innovation Institute, Virginia Tech, Blacksburg, VA, USA. [2]Department of Chemical Engineering, Ben-Gurion University of the Negev, Beer-Sheva, Israel. [3]Department of Biomedical Sciences and Pathobiology, Virginia Tech, Blacksburg, VA, USA. [4]Center for Engineered Health, Virginia Tech, Blacksburg, VA, USA. [5]Ilse Katz Institute for Nanoscale Science and Technology, Ben-Gurion University of the Negev, Beer-Sheva, Israel. [6]Present address: Department of Biomedical Engineering, Duke University, Durham, NC, USA. [7]These authors contributed equally: Yumeng Zhu, Yulia Shmidov. ✉e-mail: rbitton@bgu.ac.il; jbmatson@vt.edu

incorporated into a hydrogel. We chose the YIGSR peptide, which is derived from the ECM protein laminin, because it binds to integrins, causing them to cluster in the plasma membrane and inducing recruitment and activation of intracellular signaling molecules[27]. This concept could then be extended such that the cells themselves could trigger the switch (Fig. 1A).

## Results and discussion

### Switch peptide synthesis and verification

As the first step toward mimicking cryptic site function, we designed and synthesized (Supplementary Figs. 1–7) a non-functional depsi-peptide that could be enzymatically deprotected to reveal a functional peptide epitope. The structure of the depsipeptide is abbreviated as $KS_{YIG}R$, where the $S_{YIG}$ unit indicates a Ser residue esterified with the Ac-YIG peptide sequence (Ac = acetyl). This design effectively splits the bioactive YIGSR pentamer using an ester bond (Fig. 1B, "switch pep-tide"). Exposure to the enzyme trypsin, which cleaves peptide bonds on the C-terminal side of positively charged amino acid residues, would remove the N-terminal Lys residue ("intermediate peptide", $S_{YIG}R$), leading to transformation into a functional YIGSR cell-adhesive epitope ("functional peptide") via an O → N acyl shift.

We followed the kinetics of Lys residue cleavage by trypsin and functional peptide formation using MALDI-TOF mass spectrometry (MS). We tracked disappearance of the switch peptide peak and evo-lution of the intermediate or functional peptide peak ($S_{YIG}R$ or Ac-YIGSR; both have the same exact mass) after treatment with trypsin. MALDI-TOF MS scans (Fig. 1C) showed that the signal intensity of the

switch peptide ($m/z = 764$) started to decrease after 1 h, concurrent with the appearance of the intermediate/functional peptide peak ($m/z = 636$). After 3 h, the intensity of both signals was similar, and after 24 h the switch peptide signal disappeared completely, while the intermediate/functional peptide peak became dominant, indicating near-complete conversion.

The MALDI-TOF MS data verified that trypsin cleaves the K resi-due, but it could not confirm the O → N acyl shift because both the intermediate peptide and the functional peptide have the same molecular formula. To confirm successful rearrangement of the intermediate peptide to form the functional, bioactive YIGSR peptide, we evaluated the reaction products using both the Kaiser test, a sen-sitive colorimetric method to determine the presence of primary amines in a sample[28], and $^{13}C$-NMR spectroscopy. To avoid signals from the presence of trypsin in the samples in both experiments, we syn-thesized a modified version of the switch peptide, Fmoc-$S_{YIG}R$ (Sup-plementary Fig. 2C), which includes an Fmoc protecting group on the N-terminus of the Ser residue in place of the Lys residue. This sub-stitution allowed us to remove the Fmoc group using piperidine to generate the intermediate peptide, which rearranged into the func-tional peptide, as indicated by the yellow color of the Kaiser test solution (Supplementary Fig. 3). $^{13}C$ NMR spectroscopy also supported rearrangement: Comparing the $^{13}C$ NMR spectrum of the switch pep-tide with that of Fmoc-$S_{YIG}R$ after treatment with piperidine, we observed a change of the chemical shift for the β-carbon on the Ser residue from 63.2 ppm to 61.2 ppm (Supplementary Fig. 8). The 61.2 ppm signal matched the predicted chemical shift of the Ser β-carbon in

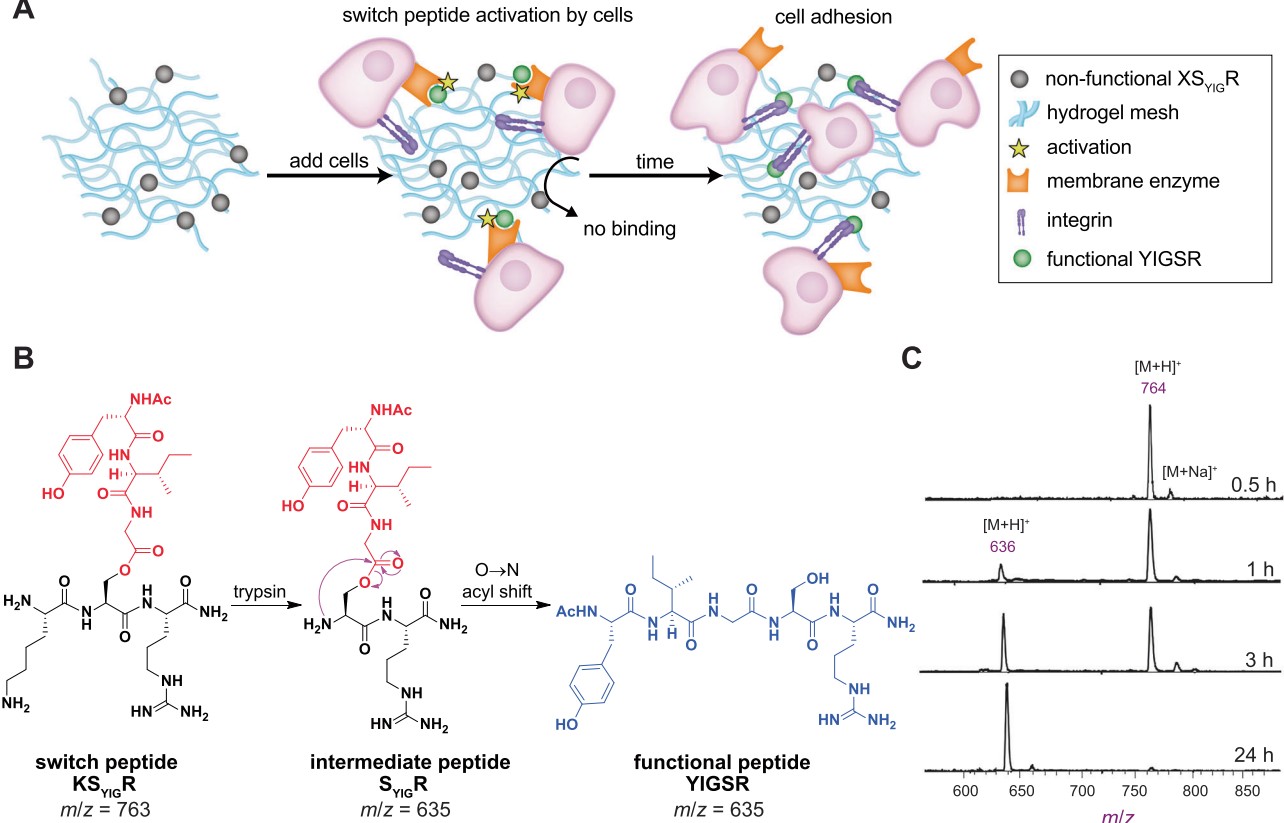

**Fig. 1 | Design of cryptic sites in a synthetic ECM using switch peptides.**
**A** Schematic illustration of synthetic ECM with switch peptides as cryptic site mimics. Cell surface enzymes remove an amino acid residue, activating the switch peptide and forming the cell-adhesive YIGSR sequence. **B** Chemical transformation of the switch peptide $KS_{YIG}R$ containing a "split sequence" where the Ser residue is attached to a side chain YIG sequence (red) through its side chain alcohol, forming an ester bond. Removal of the N-terminal Lys residue by added trypsin reveals the

free Ser amine (intermediate peptide). Once the free amine is present, a sponta-neous O → N acyl shift occurs, generating the native peptide bond and forming the functional YIGSR peptide (blue). Other examples in this report use a membrane enzyme to trigger the switch. **C** MALDI-TOF MS spectra of switch peptide after incubation with trypsin for different time points. Peak $m/z = 764$ ([M + H]$^+$) indicates the switch peptide, while $m/z = 636$ ([M + H]$^+$) indicates the functional peptide.

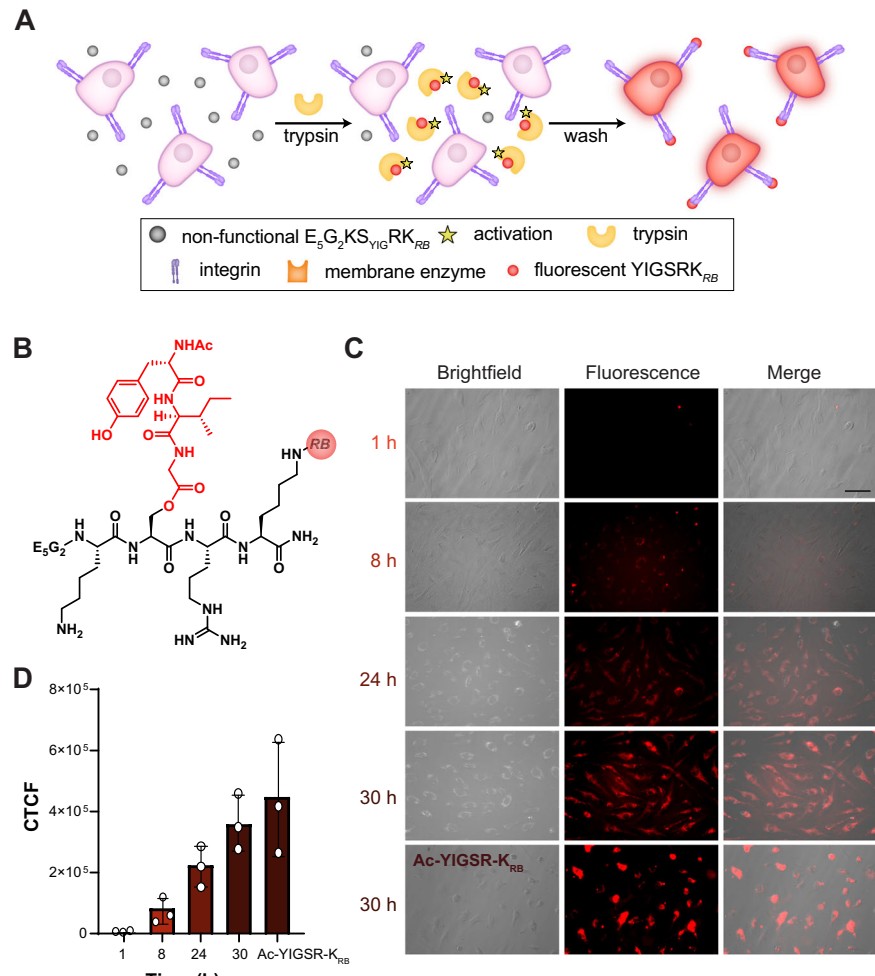

**Fig. 2 | Switch peptide rearrangement leads to gain of functionality.**
**A** Schematic illustration of HUVEC cells in culture with fluorescently labeled switch peptide ($E_5G_2KS_{YIG}R$-$K_{RB}$). Upon activation by trypsin, the peptide rearranges into a fluorescently labeled functional peptide, YIGSRK$_{RB}$, where the YIGSR epitope binds to cell surface integrins. Peptide $E_5G_2KS_{YIG}R$-$K_{RB}$ is also fluorescent, but it cannot bind to integrins and is removed in the washing step. **B** Chemical structure of fluorescently labeled switch peptide. **C** Brightfield, fluorescence, and merged images of HUVECs preincubated with Ac-$E_5G_2KS_{YIG}R$-$K_{RB}$ and trypsin at various timepoints, and positive control Ac-YIGSR-$K_{RB}$ at 30 h. Scale bar = 100 μm.
**D** Corrected total cell fluorescence (CTCF) values at each timepoint measured from fluorescence images. Statistical tests are included in Supplementary Table 1 with $n = 3$ independent experiments and 5 replicates per experiment. Data are presented as mean values ± SD.

---

the functional peptide, and the other 4 signals of the tertiary carbons in this range matched predicted chemical shifts. Taking the MS, Kaiser test, and $^{13}$C NMR spectroscopy results together, it is clear that the designed rearrangement is indeed occurring.

## Gain of biological function

We evaluated the gain of biological function by relying on the cell adhesion mechanism[29], where the YIGSR sequence binds to the cell surface laminin receptor integrin[27]. In these studies we used in vitro fluorescence cell imaging on human umbilical vein endothelial cells (HUVECs), a cell type that has been previously used to confirm that YIGSR promotes adhesion (Fig. 2A)[30]. For these experiments, we synthesized a switch peptide labeled with the fluorescent probe rhodamine B (RB) with the sequence Ac-$E_5G_2KS_{YIG}R$-$K_{RB}$ (Fig. 2B, $K_{RB}$ = Lys residue modified with rhodamine B). The design of this peptide included the $E_5G_2$ sequence on the N-terminus to increase peptide solubility, counteracting the hydrophobic $K_{RB}$ unit that was added to the C-terminus. Cell viability studies (Supplementary Fig. 9) showed no cytotoxicity up to 200 μM for peptides KS$_{YIG}$R, YIGSR, or any RB-labeled peptides.

Addition of the $K_{RB}$ unit created a new potential cleavage site for trypsin at the C-terminal side of the Arg residue (i.e., the R–$K_{RB}$ peptide bond). To ensure the desired response still occurred, i.e., cleavage of the K–$S_{YIG}$ peptide bond to remove the N-terminal Lys residue, we designed an experiment to follow the transformation of KS$_{YIG}$RK as a representative peptide that contains two potential cleavage points, the desired K–$S_{YIG}$ peptide bond and the undesired R–K peptide bond. We treated this peptide with trypsin and followed bond cleavage at these two sites using MALDI-TOF MS. The results showed that cleavage occurred at either site at approximately the same rate but not both simultaneously (Supplementary Fig. 10). This experiment suggests that for Ac-$E_5G_2KS_{YIG}RK_{RB}$, trypsin can still cleave the desired K–$S_{YIG}$ peptide bond but also the undesired R–$K_{RB}$ peptide bond. However, we envisioned that this undesired cleavage would not be a problem in cell studies because it would simply generate a $K_{RB}$ species that would be removed in washing steps.

Based on the results from the above experiment, we expected that addition of the fluorescently labeled switch peptide, Ac-$E_5G_2KS_{YIG}RK_{RB}$, and trypsin to cultured HUVECs would result in ~50% cleavage of the K–$S_{YIG}$ peptide bond (in this case trypsin removes the

entire $E_5G_2K$ sequence), inducing a rearrangement of the switch peptide into a fluorescently labeled functional peptide, $YIGSRK_{RB}$. The resulting $YIGSRK_{RB}$ sequence should then be able to attach on the cell membrane by binding to laminin receptors, resulting in cells exhibiting a strong fluorescence signal after washing away trypsin, excess peptide, and any cleaved $K_{RB}$ species (Fig. 2A). In contrast, treatment of HUVECs with the labeled switch peptide *without* trypsin should result in no binding of the switch peptides to the cell surface, leading to cells exhibiting no fluorescence signal after washing.

We treated HUVECs with switch peptide $Ac\text{-}E_5G_2KS_{YIG}RK_{RB}$ with or without trypsin at levels 2–3 orders of magnitude lower than those typically used to detach cells in cell culture. After 1 h, there was negligible fluorescent signal, consistent with the kinetics of trypsin cleavage and peptide rearrangement observed in the MALDI-TOF experiments (Fig. 1C). The fluorescence signal increased slowly over time, with a strong signal after 30 h, demonstrating that the fluorescently labeled switch peptide successfully rearranged and attached to the cell membrane. In control experiments (Supplementary Note 1), we found that cells treated with switch peptide but without trypsin showed no fluorescence signal even after incubating for 30 h (Supplementary Fig. 11 first row). Similarly, peptide $K_{YIG}RK_{RB}$, which cannot undergo a switch, showed no fluorescence with or without trypsin (Supplementary Fig. 11 second and third rows). As a positive control, we treated cells with functional peptide $Ac\text{-}YIGSR\text{-}K_{RB}$ for 30 h, which showed strong fluorescence at a level similar to that of switch peptide $Ac\text{-}E_5G_2KS_{YIG}RK_{RB}$ treated with trypsin (Supplementary Fig. 11 fourth row). Quantification of fluorescence micrographs for each treatment group (Fig. 2D) validated our conclusions. Finally, we also fixed the cells and labeled cell surface receptor CD29 (integrin β1, a laminin receptor), observing co-localization of the integrin stain with the rhodamine B-labeled peptide (Supplementary Fig. 12), confirming binding to integrin β1.

## Revealing cryptic sites in a synthetic ECM

Following the successful shift from a non-functional epitope to a functional one in solution, we examined our hypothesis that switch peptides can mimic cryptic sites within synthetic ECMs. We covalently attached the switch peptide to alginate to create a switch peptide-modified alginate, which formed a hydrogel upon addition of $CaCl_2$ (Fig. 3A, B). Alginate hydrogels (physically crosslinked with $Ca^{2+}$) are widely used as synthetic ECM materials[31]. Their stiffness and porosity can be varied to allow proper mass transport and cell support. For hydrogels used in this study, enzymatic mobility is not likely to be significantly reduced, as can be expected from the large hydrogel pore size, which according to the SEM image (Fig. 3B) is on the order of hundreds of nm, while still maintaining sufficient stiffness to provide cell support (Supplementary Fig. 13). Upon exposure to trypsin, we envisioned that this hydrogel would transform into a YIGSR-modified alginate hydrogel capable of cell adhesion (Fig. 3C). We chose alginate for this purpose because it has been extensively explored as a synthetic ECM, primarily due to its inertness and ease of crosslinking using calcium ions[32]. Alginate itself lacks the ability to form specific cellular interactions, thus any change in cell behavior following trypsin activity can be attributed to exposure to synthetic cryptic sites, specifically the YIGSR functional peptide epitope.

To conjugate the switch peptide to an alginate backbone, we utilized carbodiimide chemistry, in which an amine group from the peptide reacts with alginate carboxyl groups to form an amide bond, a method widely used in alginate modification (Supplementary Fig. 14)[31]. To accomplish this, we synthesized switch peptide $Ac\text{-}KS_{YIG}RK$, which included an additional C-terminal Lys residue that could bind to alginate via its free ε-amine without impacting the switch process. Although the N-terminal Lys ε-amine could also react with alginate, after attachment this peptide wouldn't be cleaved upon addition of trypsin. Thus, about half of the $Ac\text{-}KS_{YIG}RK$ should bind to alginate with

the correct regiochemistry to enable a switch upon addition of trypsin. We also prepared an authentic functional peptide Ac-YIGSRK for control studies. Cell viability studies (Supplementary Fig. 9) showed no cytotoxicity up to 200 μM for peptides $KS_{YIG}RK$, Ac-YIGSRK, $KS_{YIG}RK$, trypsin, and 50 μM curcumin.

Conjugation of the $Ac\text{-}KS_{YIG}RK$ switch peptide to the alginate backbone was verified using UV-Vis (Supplementary Fig. 15) and $^1$H-NMR (Supplementary Fig. 16) spectroscopy. The $^1$H NMR spectrum of the lyophilized product showed the expected signals consistent with peptide attachment, and the coupling efficiency was estimated to be 58% by UV-vis spectroscopy (Supplementary Fig. 15C). Alginate-peptide hydrogels were physically crosslinked by addition of $CaCl_2$ solution[33]. To characterize the cellular response to the hydrogels, we measured the ability of switch peptide-modified alginate hydrogels to promote cellular adhesion of HUVECs (Fig. 3C). Cells were seeded on top of pre-formed switch peptide-modified hydrogels, with pristine (unmodified) hydrogels and modified hydrogels without trypsin as negative controls, and functional peptide-modified hydrogels as a positive control. Trypsin at a concentration nearly 100-fold lower than what is typically used for detaching cells was added to the hydrogels just before seeding the cells. Cells were incubated for either 24 or 72 h, then treated with live/dead markers and imaged to assess viability and morphology.

Figure 3D displays the fluorescence images of the hydrogels after 72 h (images after 24 h are shown in Supplementary Fig. 17). No cells were present in the pristine hydrogels (Fig. 3D, row 1), indicating that the cells did not adhere to the inert alginate hydrogel and were therefore removed during washing step. For the switch peptide-modified hydrogels that were not treated with trypsin, some cells attached to the hydrogel (Fig. 3D, row 2), probably due to the exposed cationic ε-amine of the switch peptide N-terminal Lys residue as well as the cationic Arg residue. However, this non-specific adhesion did not promote spreading of the cells, as suggested by their morphology (small and rounded) and the high number of dead cells. Cells cultured on functional peptide-modified hydrogels (positive control) spread well on the hydrogel surface and remained mostly alive (Fig. 3D, row 3). For the switch peptide-modified alginate hydrogels treated with trypsin, similar results were observed (Fig. 3D, row 4). These hydrogels promoted a high number of live cells spread across the surface, consistent with cell adhesion onto the hydrogel scaffold. Quantitative cell imaging data (Supplementary Fig. 17D) highlight that the switch peptide promotes adhesion as effectively as the functional peptide-modified hydrogel. Overall, these results demonstrate that enzyme-triggered primary sequence rearrangement transforms a poorly adhesive hydrogel into a strongly adhesive one suitable for cell attachment and growth by revealing the hidden YIGSR epitope.

## Cells reveal cryptic sites

In native tissue, enzymes expressed by cells initiate changes in the primary sequences of ECM proteins, revealing cryptic sites. To mimic this process, we next aimed to enable cell-triggered primary sequence rearrangement utilizing the enzyme aminopeptidase N (APN, also known as CD13)[34], which is expressed by endothelial cells. APN is a membrane-bound ectoenzyme that plays crucial roles in regulating various cell functions, including cell migration, invasion, angiogenesis and metastasis of tumor cells[35]. It cleaves neutral, N-terminal amino acids from peptide chain ends, with Ala being the most favored residue[36]. Thus, we designed another switch peptide with the sequence $AS_{YIG}RK$ (Supplementary Fig. 7C), replacing the trypsin-cleavable Lys residue in our original design with the APN-cleavable Ala residue.

We first confirmed APN activity in HUVECs by treating the cells with the APN substrate L-alanine-4-methyl-7-coumarinylamine (Ala-MCA). Ala-MCA showed negligible fluorescence itself, but in the presence of HUVECs, strongly fluorescent 7-amino-4-methylcoumarin

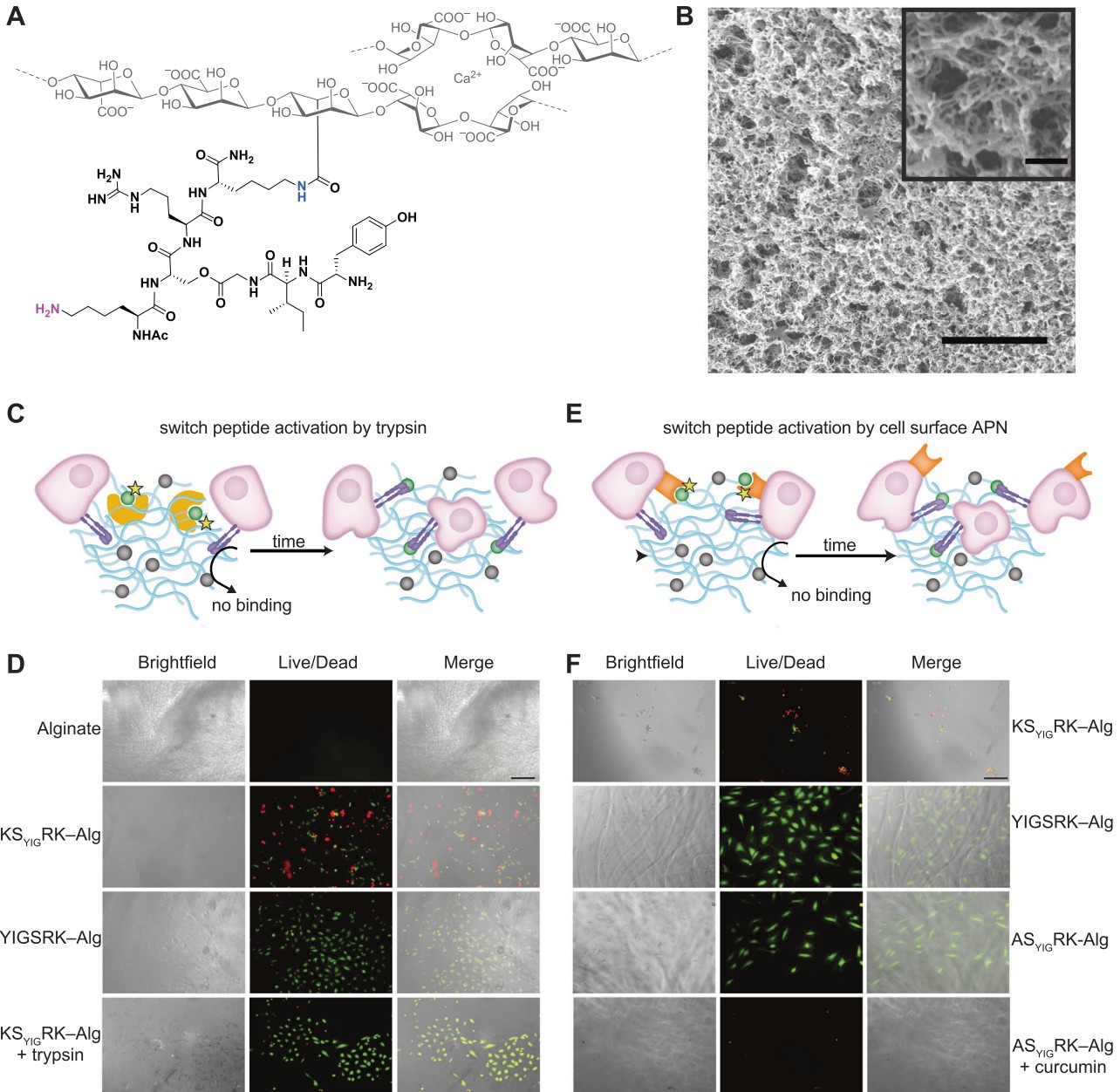

**Fig. 3 | Revealing cryptic sites in a synthetic ECM leads to gain of biofunctionality. A** Chemical structure of alginate–switch peptide hydrogel, gelled with Ca²⁺. An amide bond links the ε-amine of the C-terminal Lys (blue) residue with the alginate backbone, while the ε-amine of the N-terminal Lys residue (pink) remains free. **B** SEM images of switch peptide-functionalized alginate hydrogel. Scalebar = 2 µm (200 nm for inset). **C** Schematic illustration describing gain of biofunctionality in the synthetic ECM activated by addition of trypsin. Trypsin cleaves the N-terminal Lys residue of the switch peptide, leading to rearrangement into a functional YIGSR sequence. **D** Representative brightfield, live/dead (green/red), and merged images of HUVEC cells after 72 h on unmodified alginate

hydrogel, switch peptide-modified alginate hydrogel, functional peptide-modified alginate hydrogel, and switch peptide-modified alginate hydrogel with 0.003 wt.% trypsin added. Scale bar represents 200 µm. **E** Schematic illustration describing gain of biofunctionality in the synthetic ECM activated by cell surface APN enzymes. APN cleaves the N-terminal Ala residue of the switch peptide, leading to rearrangement into a functional YIGSR sequence. **F** Representative brightfield, live/dead (green/red), and merged images of HUVEC cells after 72 h on KS_{YIG}RK switch peptide-modified alginate hydrogel, functional peptide-modified alginate hydrogel, AS_{YIG}RK-modified alginate hydrogel, and AS_{YIG}RK-modified alginate hydrogel with curcumin (inhibitor, 50 µM). Scale bar represents 100 µm.

(AMC) was released, confirming APN activity. We also showed that APN in HUVECs could be inhibited using 50 µM curcumin, based on the reported IC50 value of 10 µM[37], substantially decreasing fluorescence intensity (Supplementary Fig. 18). We also found that curcumin was not toxic to HUVECs at this level (Supplementary Fig. 10C, 19).

After confirming sufficient APN activity in HUVECs, we then evaluated whether the non-functional peptide AS_{YIG}RK could be triggered by HUVECs to transform a poorly adhesive hydrogel into one suitable for cell attachment (Fig. 3E). We attached the AS_{YIG}RK peptide

to alginate and monitored cell adhesion on this hydrogel and various controls, then treated the cells with live/dead markers to assess their viability and morphology. No external enzymes or other triggers were added to initiate the peptide sequence switch. After 72 h incubation (see 24 h images in Supplementary Fig. 19), no cells adhered to pristine alginate (negative control, Supplementary Fig. 19C first row). A hydrogel modified with the KS_{YIG}RK sequence, which cannot be cleaved by APN, showed only a few live but rounded cells and many dead cells (Fig. 3F, first row). Cells adhered well to hydrogels modified

with the functional YIGSR peptide (positive control, Fig. 3F, second row), as in the previous experiments.

Finally, we tested the ability of HUVECs to trigger a primary sequence switch through their cell-surface APN enzymes. As predicted, switch peptide-modified alginate hydrogels with the APN-cleavable AS$_{YIG}$RK sequence promoted cell viability and spreading across the hydrogel surface (Fig. 3F, third row). Furthermore, curcumin effectively inhibited APN activity, preventing Ala cleavage and the peptide switch; thus, there were no live cells observed on these hydrogels (Fig. 3F, fourth row). The use of this inhibitor thus provides a simple off-switch that prevents peptide rearrangement. Quantitative cell imaging data (Supplementary Fig. 19D) supported our qualitative conclusions, indicating that the AS$_{YIG}$RK-modified alginate hydrogel promoted cell spreading and adhesion of HUVECs through their own enzymatic activity, mimicking native cryptic sites in this proof-of-concept system. We note that while esterases or lipases may slowly hydrolyze the depsipeptide ester bond, these cell studies suggest that any such reactions are substantially slower than the APN-mediated cleavage reaction because ester cleavage would result in a non-functional ASRK peptide.

In summary, by employing a synthetic approach involving depsipeptide bonds as linkages capable of undergoing triggered primary sequence rearrangement, we demonstrate transformation of an inert material into a synthetic ECM. This switch peptide-functionalized alginate hydrogel system offers a platform that mimics cryptic sites, which participate in the interplay between cells and native ECM. These findings show that through chemical design, a synthetic ECM can gain biofunctionality upon enzymatic action. Potential applications include tissue engineering and regenerative medicine approaches that enable cells and their synthetic ECM environment to change over time, for example during stem cell differentiation processes, where specific signals are often required at certain times in the differentiation process. Moving forward, the switch peptide approach may enable various depsipeptides to rearrange into a wide variety of functional epitopes; other specific stimuli in addition to enzymes can also be envisioned including changes in pH or the presence of certain metabolites capable of cleaving a specific bond. Elevated enzyme activities and high levels of reactive metabolites are often present in aberrant microenvironments; thus, these tools and approaches provide the potential for precise theranostic biomaterial designs that mimic critical dynamic aspects of native ECM.

## Methods
### Materials
2-(N-Morpholino)ethanesulfonic acid (MES buffer), N-hydroxysuccinimide (NHS), N-(3-dimethylaminopropyl)-N′-ethylcarbodiimide hydrochloride (EDC), and a Kaiser test kit were purchased from Sigma-Aldrich (St. Louis, MO, USA). Trypsin (0.25%) − EDTA was purchased from VWR (Radnor, PA, USA). N,N′-Diisopropylcarbodiimide (DIC), N,N-diisopropylethylamine, and hexafluoroisopropanol (HFIP) were purchased from Chem-Impex (Wood Dale, IL, USA). 4-Dimethylaminopyridine (DMAP) was purchased from Matrix Scientific (Columbia, SC, USA). All Fmoc amino acids, hexafluorophosphate benzotriazole tetramethyl uranium (HBTU), 2-chlorotrityl chloride resin, and Rink amide MBHA resin were purchased from Peak Polypeptide Biosystems LLC (P3 BioSystems, Louisville, KY, USA). Piperidine, Rhodamine B, and acetic anhydride were purchased from Beantown Chemical (Hudson, NH, USA). Alginic acid sodium salt ($M_w$ = 300 kDa) was purchased from Research Products International (RPI, Mt. Prospect, IL, USA). Human umbilical vein endothelial cells (HUVECs) were a kind gift from Professor Padma Rajagopalan (Department of Chemical Engineering, Virginia Tech, Blacksburg, VA, USA). Human EC Growth Medium was purchased from Cell Applications, Inc. (San Diego, CA, USA). Dulbecco's Phosphate-Buffered Saline (DPBS) was purchased from Gibco (Thermo Fisher Scientific,

Inc., Waltham, MA, USA). Fetal bovine serum (FBS) was purchased from VWR (Radnor, PA, USA). 50 IU/mL penicillin and 50 µg/mL streptomycin were purchased from MP Biomedicals (Santa Ana, California, USA). All other reagents were obtained from Sigma-Aldrich (St. Louis, MO, USA) or VWR (Radnor, PA, USA), unless otherwise stated.

### Peptide synthesis and purification
Peptides were synthesized either manually or using a Liberty 1 microwave-assisted peptide synthesizer (CEM) using solid-phase peptide synthesis (SPPS) via standard fluorenylmethoxycarbonyl (Fmoc) protocols as described previously[38,39]. The linear tripeptide Fmoc-Lys(Boc)-Ser-Arg(Pbf) (Fmoc-KSR, Supplementary Fig. 2A) was first synthesized by HBTU and diisopropylethylamine (DIEA) coupling methods on MBHA Rink resin. Separately, the side-chain peptide Ac-Tyr-Ile-Gly-OH (Ac-YIG-OH, Supplementary Fig. 2B) was prepared on 2-chlorotrityl chloride resin, then cleaved from the resin using HFIP and precipitated from Et$_2$O. An esterification reaction between the Ser hydroxyl group on resin-bound Fmoc-KSR and the Gly carboxylic acid in Ac-YIG-OH was performed using DIC/DMAP as follows: Ac-YIG-OH (0.3 mmol) and DMAP (0.01 mmol) were dissolved in tetrahydrofuran (THF) and then added to Fmoc-KSR-resin (0.1 mmol), followed by the addition of DIC (0.3 mmol). After the mixture was stirred at room temperature for 2 h, fresh Ac-YIG-OH, DIC, and DMAP were added and stirred overnight to ensure full conversion of the coupling. The Fmoc group was then removed by addition of 10 mL piperidine (20% v/v in DMF) for 15 min (two times). N-terminal acetylation was achieved by treating the resin with 20% acetic anhydride in dimethylformamide (DMF) (v/v) with 100 µL DIEA three times (10 mL each). After cleavage and isolation, the crude KS$_{YIG}$R peptide was dissolved in water containing 0.1% trifluoroacetic acid (TFA) and filtered through a 0.45 µm polytetrafluoroethylene (PTFE) filter before purification by preparative HPLC (described below). KS$_{YIG}$RK, AS$_{YIG}$RK, and YIGSRK (Supplementary Fig. 7) were synthesized by the same procedures as mentioned above.

All peptides were purified by preparative RP-HPLC using an Agilent Technologies 1260 Infinity HPLC system (Agilent Technologies, Santa Clara, CA) equipped with a fraction collector. Separations were performed using an Agilent PLRP-S column (100 Å, 10 µm, 150 × 25 mm) monitoring at 220 nm. The expected mass was confirmed using ESI-MS (Advion ExpressIon Compact Mass Spectrometer) and matrix-assisted laser desorption/ionization time-of-flight mass spectrometry (MALDI-TOF MS). Fractions containing pure products were combined and lyophilized (FreeZone −105 °C, Labconco, Kansas City, MO), and then stored at −20 °C until needed.

### Mass spectrometry (MS)
Switch peptide solution (100 µL, 5 mg/mL in PBS) was mixed with 20 µL of a trypsin solution (0.25%) in a vial and placed at 37 °C in an incubator. Aliquots (2 µL) from each reaction solution were withdrawn at specific times, and trypsin was deactivated immediately by the addition of acetic acid (0.3 µL, 1 M in distilled water). Samples (0.8 µL) were then mixed with a matrix solution (0.8 µL, 0.02 M α-cyano-4-hydroxycinnamic acid dissolved in acetonitrile/methanol/water (1:1:1) with 0.1% formic acid) and deposited on a MALDI plate (Opti-TOF 384 Well, 123 × 81 mm). Samples were analyzed using a matrix-assisted laser desorption ionization−tandem time-of-flight mass spectrometer (4800 MALDI TOF/TOF, AB Sciex).

### Nuclear magnetic resonance (NMR)
NMR spectra were measured on Agilent 500 MHz spectrometers at 22 °C. $^1$H and $^{13}$C NMR chemical shifts are reported in ppm relative to internal solvent resonances. Unless otherise noted, 20−30 mg samples were dissolved in deuterium oxide (D$_2$O) or dimethyl sulfoxide-$d_6$ (DMSO-$d_6$).

## Cell culture

Cell studies were conducted using human umbilical vein endothelial cells (HUVECs) at passages 5–12. Cultures were grown in endothelial cell growth medium (ECGM) supplemented with 10% fetal bovine serum (FBS), 50 IU/mL penicillin, and 50 µg/mL streptomycin (MP Biomedicals). Cells were cultured at 37 °C in 5% $CO_2$-air. The media was changed every other day. The cultures were passaged after 70–80% confluence was achieved. Cells were rinsed with 1X PBS solution three times, and then released with 0.05% trypsin-EDTA solution (VWR, Radnor, PA). The suspension of released cells was centrifuged at $200 \times g$ (1000 rpm) for 5 min before counting and plating for experiments.

## Cell viability assays

HUVECs (ATCC catalog number PCS-100-010) were plated in a 96-well plate at a density of 5000 cells per well in 200 µL complete ECGM per well. After culturing for 24 h, the media was discarded and the cells were washed with 1X PBS three times before 190 µL serum-containing ECGM media was added. Next, varying amounts of peptide stock solution (4 mM in PBS) were diluted to 100 µL with PBS and added to the wells to make the final peptide concentrations ranging from 10 to 200 µM. After incubation for 24 h, cells were washed three times with PBS and then treated with serum-free ECGM media (100 µL). Next, 10 µL cell counting kit 8 solution (CCK-8, Dojindo, Rockville, MD) was added to each well. After incubation for another 3 h to allow for development of the CCK8 dye, absorbance was recorded at 450 and 750 nm using a BioTek Synergy Mx plate reader (BioTek, Winooski, VT). Worked-up data (absorbance at 750 nm subtracted from absorbance at 450 nm) were graphed using GraphPad InStat, version 3 (GraphPad Software, Inc., San Diego, CA). Mean values are reported together with the standard deviation (SD) representing the combination of 3 independent experimental runs with five replicates per experiment.

## In vitro studies of biological function using fluorescent probe

HUVECs were plated at a density of 0.1 million cells per well in a 24-well plate with 0.5 mL complete ECGM per well and cultured for 24 h, followed by washing with 1× PBS three times and the addition of 950 µL serum-free ECGM per well. Rhodamine B labeled peptide solution (50 µL in serum-free ECGM, 1 mM) was mixed with 0.05% trypsin (5 µL). The mixed solution was then added to each well for a total volume of 1005 µL. Cells were incubated with the mixed solution for various time periods. At different time points (1 h up to 30 h), the media was removed, and cells were washed with PBS three times. Fresh PBS (500 µL) was added to the wells, and the plate was visualized by brightfield and fluorescence microscopy (Nikon Eclipse Ti–U) with TRITC filter set. The magnification was 20×. Corrected total cell fluorescence (CTCF) intensity was estimated using the equation CTCF = integrated density − (area of selected cell) × (mean fluorescence of background). A rectangular drawing/selection tool in ImageJ was used to select a desired region with fluorescence, and its integrated density and area were measured. Mean fluorescence intensity of the background was calculated by averaging the intensities of selected areas that had no fluorescence. Cell counts were 30 for each group from three separate wells.

## Preparation of alginate-switch/functional peptide conjugates

Conjugation of the switch peptide as well as the functional peptide to an alginate backbone was performed using carbodiimide chemistry[40,41]. Alginate was dissolved in MES buffer (0.1 M, pH 6.0) at a concentration of 10 mg/mL and stirred overnight to give a 1% w/v aqueous solution. NHS (0.16 mg per mL alginate solution) and EDC (0.50 mg per mL alginate solution) were added as solids to afford a molar ratio of NHS:EDC of 1:2 to activate approximately 5% of alginate carboxylic acid groups. After stirring for 5 min, peptide (0.33 mg per

mL alginate solution) was added as a solid, and the solution was then stirred continuously overnight. Thereafter, the solution was dialyzed against triple distilled water in 6k-8k Da MWCO dialysis tubes (Spectrum™, Thermo Fisher Scientific, Inc., Waltham, MA, USA) for 48 h and then lyophilized. For analysis, the required amount of the dried conjugated polymer was dissolved in triple distilled water.

## Preparation of alginate-switch/functional peptide gels

Vials containing lyophilized alginate-peptide conjugates were sprayed with 70% alcohol and sterilized inside a biosafety hood under UV for at least 2 h, followed by dissolving in sterilized water in the biosafety hood. 150 µL of 1% w/v different alginate stock solutions (alginate, alginate-switch peptide, alginate-functional peptide) were added to wells in a 48-well plate, followed by the addition of 20 µL $CaCl_2$ solution (100 mM in DI water) into the center of each well. Samples were prepared in triplicate and allowed to gel overnight in the biosafety hood after covering the well plate. The formed gels were taken out of the wells with small tweezers and carefully placed in new wells in a new 48-well plate and washed with 200 µL 1X PBS solution three times to remove excess $CaCl_2$. After washing, the formed gels were transferred carefully with tweezers into surface-uncoated (untreated) 96-well plates, with the gels fully covering the bottom of each well.

## Scanning electron microscopy (SEM)

SEM images were taken on a FEI Verios field emission scanning electron microscope. Gel samples were prepared by critical point drying (CPD) of swollen gels as follows: the excess specimen media was removed, and a fixative solution (2% glutaraldehyde + 3% sucrose in triply distilled water) was added and left overnight. Dehydration in the presence of ethanol was performed by rinsing the gels for 15 min in rising concentrations of ethanol in water in the following order: 20%, 50%, 70%, 80%, 90%, 95%, and 100%. Ethanol exchange with $CO_2$ was performed by a CPD instrument (Tousimis 931. GL). Samples were then coated with a few nm thick carbon coating (Emitech K575X) before imaging.

## Rheology

The viscoelastic properties of the hydrogel samples were determined by oscillatory rheology experiments. Measurements ($n = 3$) were carried out using an MCR 302 rheometer (Anton Paar, Rhenium) equipped with stainless steel parallel plates ($d = 14$ mm). First, strain sweep experiments (frequency of 10 rad s$^{-1}$) were performed to establish the linear viscoelastic region of the hydrogels, and a strain of 0.8% was chosen for frequency sweep measurements. Hydrogels were prepared in a 24-well plate from lyophilized alginate-switch peptide conjugates dissolved in sterilized water. Next, 300 µL of 1% w/v alginate-switch peptide stock solutions were added to the wells, followed by the addition of 40 µL $CaCl_2$ solution (100 mM in DI water) into the center of each well. Samples were prepared in triplicate and allowed to gel overnight.

## In vitro studies of cell adhesion on alginate-peptide hydrogels

Trypsin (0.05%, 10 µL) was added to the wells containing alginate-switch peptide gel in a 96-well plate before seeding the cells. HUVEC cells were seeded on the top of the hydrogels at a density of 10,000 cells per gel with 150 µL serum-free ECGM media. Cells were cultured in an incubator for 24 h or 72 h. After each incubation time, the media was removed and 150 µL fresh serum-free ECGM media was added to each well. The live/dead combined solution was prepared by adding 2 µL of 2 mM EthD-1 stock solution (Component B) to 1 mL D-PBS. The solution was vortexed, and then 0.5 µL of 4 mM calcein AM stock solution (Component A) was added to this solution to give a final working solution (2 µM calcein AM and 4 µM EthD-1). The live/dead working solution was vortexed to mix well, and 20 µL of the solution was added to each well. Cells were incubated for 45 min. The gels were

then visualized by bright-field and fluorescence microscopy (Nikon Eclipse Ti−U) with FITC and TRITC filter set. The magnification was 10× and 4×. Corrected total cell fluorescence (CTCF) intensity was estimated using the equation CTCF = integrated density − (area of selected cell) × (mean fluorescence of background). Cell counts were 30 for each group from three separate wells. A rectangular drawing/selection tool in ImageJ was used to select a desired region with fluorescence, and its integrated density and area were measured. Mean fluorescence intensity of the background was calculated by averaging the intensities of the selected areas that had no fluorescence.

**In vitro aminopeptidase N (APN) activity assay**
HUVEC cells were plated in a 96-well plate at a density of 5000 cells per well in 200 μL complete ECGM V2 media per well. After culturing for 24 h, cells were washed with 1X PBS two times. The cells were then treated with 100 μM L-alanine-4-methyl-7-coumarinylamine trifluoroacetate (Ala-MCA) in PBS with or without inhibitor curcumin (50 μM in DMSO) for 24 h in an incubator (37 °C, 5% $CO_2$). The release of fluorescent product 7-amino-4-methylcoumarin in the media was monitored on the plate reader (BioTek Cytation3 imaging reader) with $\lambda_{ex}$ of 360 nm and peak $\lambda_{em}$ of 460 nm.

**In vitro studies on cell-triggered exposure of cryptic sites**
Hydrogels were synthesized as described in the protocol mentioned above with all the peptides: $AS_{YIG}RK$, $KS_{YIG}RK$, and YIGSRK. Hydrogels were then loaded into an untreated 96-well plate, ensuring that the bottom of the plate was completely covered. HUVEC cells were seeded on the top of the hydrogels at a density of 10,000 cells per gel with 150 μL complete ECGM media and cultured for 24 h or 72 h. After each incubation period, the media was removed and 150 μL fresh ECGM media was added to each well. The live/dead images were obtained as described above. The cells were incubated for 45 min at 37 °C in 5% $CO_2$-air. The gels were then visualized by bright-field and fluorescence microscopy (Nikon Eclipse Ti−U) with FITC and TRITC filter set. The magnification was 10× and 4×. Corrected total cell fluorescence (CTCF) intensity was estimated using the equation CTCF = integrated density − (area of selected cell) × (mean fluorescence of background). A rectangular drawing/selection tool in ImageJ was used to select a desired region with fluorescence, and its integrated density and area were measured. Mean fluorescence intensity of the background was calculated by averaging the intensities of the selected areas that had no fluorescence.

**Immunohistochemistry**
HUVEC cells were seeded in an 8-chamber glass slide (Lab-Tek Chamber Slide) at a density of 30,000 cells per well in 0.4 mL complete ECGM media per well. After culturing for 24 h, cells were washed twice with 1× PBS. The cells were then incubated with 395 μL of serum-free ECGM media for 3 h in the following groups: Blank (5 μL PBS), and 4 mM Ac-YIGSR-$K_{RB}$ (5 μL in PBS, final concentration 50 μM). After incubation for 24 h, the media was removed, and cells were washed with PBS three times. Cultured cells were then fixed in 10% formalin for 5 min, then washed three times with PBS for 5 min each. Slides were blocked for 1 h with 2% cold water fish skin gelatin (Sigma, Inc., St. Louis, MO, USA) and 0.2% triton-X100 solution. Slides were then incubated in conjugated primary antibody diluted in blocking solution overnight at room temperature, then washed with PBS five times for 10 min each. The antibody used was FITC Armenian hamster anti-mouse/rat CD29 at a 1:50 dilution (Biolegend, San Diego, CA, USA; Catalog number 102205, clone HMbeta1-1), validated by the manufacturer by immunofluorescent staining with flow cytometric analysis. Finally, slides were mounted with DAPI Fluoromount-G (SouthernBiotech, Birmingham, AL, USA). Images were then obtained using a Nikon A1 Confocal Microscope with NIS Elements Imaging Software (Melville, New York, USA).

**Statistics and reproducibility**
Cell experiments in Figs. 2C and D, 3D and F, Supplementary Figs. 9, 11, 17, and 19 were all repeated three times with similar results. SEM (Fig. 3B) and confocal imaging (Supplementary Fig. 12) were done once. Group comparisons are indicated as determined by a one-way analysis of variance (ANOVA) with a Student−Newman−Keuls comparisons post hoc test.

**Reporting summary**
Further information on research design is available in the Nature Portfolio Reporting Summary linked to this article.

## Data availability
Source data are provided with this paper.

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

## Acknowledgements
This work was supported by the Binational Science Foundation (2016096 to J.B.M. and R.B.) and the National Institutes of Health (R01GM123508 to J.B.M.). We thank Dr. Matthew Webber, Dr. Yin Wang, Zhao Li, and Dr. Mingjun Zhou for helpful discussions; Katlyn F. Morales for experimental assistance; Dr. Padma Rajagopalan and Rosalyn Hatlen for HUVEC cells and providing suggestions on cell culture; Dr. Andrew Lowell and Dr. Jennifer McCord for instrument access; Gal Yosefi for SEM; Dr. Olga Iliashevsky for assistance with hydrogel characterization; Dina Aranovich for helpful suggestions on cell studies; and Dr. Keith Ray for assistance with MALDI-TOF experiments.

## Author contributions
Y.Z., Y.S., E.A.H., and M.H.T.: formal analysis, investigation, methodology, validation, visualization, writing – original draft, and writing – reviewing and editing. R.B. and J.B.M.: conceptualization, funding acquisition, supervision, visualization, writing – original draft, writing – reviewing and editing.

## Competing interests
The authors declare no competing interests.
