## [Peer Review File · Nature Communications]

REVIEWER COMMENTS

Reviewer #1 (Remarks to the Author):

Matson and collaborators report the synthesis and in vitro evaluation of cryptic sites in synthetic extracellular mimics (ECMs). From a chemistry perspective the cryptic sites rely on protected depsipeptides, which upon enzymatic cleavage release a free N-terminal serine or threonine ester which leads to O->N acyl shifts under the correct pH value. The manuscript is very well structured and demonstrates in increasing order of complexity the N-terminal cleavage of lysine to induce the O->N acyl shift by trypsin, both in buffer and in presence of HUVEC cells. Starting with the K extended depsipeptide KSYIGR to release YIGSR, or fluorescent labelled Ac-E5G2KSYIGRKRK, the next step was to evaluate cell viability and adhesion using alginate materials which are functionalized with cryptic sites or caged adhesion epitopes Ac KSYIGRK. Finally, the final and most convincing experiment is discussed using triggered peptide sequence rearrangement based on cellular enzymes which do not need to be added externally. The group uses the enzyme aminopeptidase N (APN) which is expressed by endothelial cells, in this case the HUVEC cell lines mentioned already. To achieve this ASYIGRK sequence was used where the trypsin-cleavable Lys residue was substituted with an Ala residue. Quantitative cell imaging was used to support the conclusion that ASYIGRK-modified alginate hydrogel materials promote cell spreading and adhesion of HUVECs relying on their own enzyme machinery.

The manuscript reads very well. It is an elegant approach to embed synthetic cryptic sites in ECM materials, and combines all the strengths of synthetic macromolecular chemistry with applications in biomaterials sciences. The experimental data is very convincing and fully support the main conclusions of the manuscript. It will appeal to the broad readership of Nature Communications. Publication with minor revision is recommended, if the following comments are addressed:

- Physical crosslinking of alginate biopolymers to yield hydrogel materials is performed with addition of CaCl₂. Can authors add few words about the limit of CaCl₂ addition, which would still produce sufficiently stiff viscoelastic materials (please add some data for storage and loss moduli) but at the same time not reduce enzymatic activity and mobility.
- Looking at some of the advanced protected and labeled probes with in situ labelling of cellular integrins, it seems that a fair bit of molecular engineering was required. Can the authors comment on whether it was just solubility that was improved with added negative charged using the 5 glutamic acids

in Ac-E5G2KSYIGRKRKB, or were they also needed to avoid unspecific binding on the cells or even cell uptake?

- Lipases can also be present in extracellular environments. At which point are O->N acyl shifts outcompeted with detrimental ester hydrolysis? Did authors observe any side reactions, and could a comment be added to the manuscript?

Reviewer #2 (Remarks to the Author):

The manuscript by Zhu et al., entitled “Activating Hidden Signals: Mimicking Cryptic Sites in a Synthetic Extracellular Matrix”, describes “switch peptides” which can be activate by trypsin/ enzyme aminopeptidase N (APN) then change into bioactive adhesive pentapeptide YIGSR. The authors modify the “switch peptides” with rhodamine B, then use rhodamine B labeled peptide and low concentration trypsin to treat the HUVECs for different times, show the switch peptides can switch to YIGSR and combine with HUVECs after co-cultrul for 30 hours. Then they conjugate the “switch peptides” with alginate to make a hydrogel, only treat with trypsin/APN then the HUVECs can live on the alginate–switch peptide hydrogel. The authors use a smart design to show the HUVECs can identify the cryptic sites in a synthetic Extracellular Matrix and attach to it. However, the authors should consider the following comments to improve their manuscript.

1. The authors states “Activating Hidden Signals” in the title of the manuscript, but they don` t have any evidences to show any signal pathway had been activated in the manuscript, maybe the authors should re-consider the manuscript title.

2. The authors could consider to label some cell structure protein such as Focal Adhesion Kinase (FAK) or integrin when use the rhodamine B labeled peptide to treat the HUVEC to show the peptides are specific combined with cell structure and activate some signal pathway.

3. The authors should state in Fig 3, Fig S12 and Fig S14, which color represents living cells, which represents dead cells.

4. The authors should make statistics test to check if there are significant differences in fig 2D, fig S8B, fig S12D and fig S14D.

5. The authors should improve the image's quality. It is difficult to identify the cells in the Fig 3D row 2 brightfield image, Fig 2D row 2 brightfield image, while there are strong fluorescence signals in the relative fluorescence images.

6. The manuscript writing is a little confusion. The extended fig S3, fig S7 and fig S10 are not mentioned in the main text. The fig S6 is mentioned in line 102, while the fig S5 is mentioned in line 137 and fig S4 is mentioned in line 251, it is difficult to read the results.

Reviewer #3 (Remarks to the Author):

The manuscript by Zhu et al, entitled "Activating Hidden Signals: Mimicking Cryptic Sites in a Synthetic Extracellular Matrix", present a nice interdisciplinary work at the interface between Chemistry and Cell Biology. Ultimately, this work offers a very original strategy to engineer a synthetic ECM that can expose cryptic sites upon enzymatic cleavages, either by experimental addition of a soluble enzyme (trypsin) or by an enzyme naturally present at cell surface (aminopeptidase N). I appreciate the efforts to improve the understanding of the complex experiments by addition of several schematic illustrations. I am happy to support the publication of this manuscript in Nat Comm journal, provided that the authors address the points listed below.

I have one very major concern that should be experimentally addressed. Curcumin was shown to be very toxic on many cell types, 50 microM is a very high concentration. The important results of Fig. 3F, fourth row, may be explained by the toxicity of the curcumin, instead of by its inhibition of aminopeptidase N (APN) activity. An important experiment is lacking: the adhesion of HUVEC to YIGSRK-Alg in presence of 50 microM curcumin. If curcumin toxicity may turn out to be a problem, the authors could try to knock-down by siRNA the CD13 gene expressing APN, or to compare cell lines with different expression levels/activities of APN.

Other specific points:

I strongly recommend to move to main figures some data that are now presented in extended data section, since they are crucial information. Move Fig. S8 to Fig. 2. Move quantification of Fig. S12D and Fig. S14D to Fig. 3.

How many times were reproduced the experiments presented in Fig.2 and Fig.3?

Please, explain better the CTCF quantifier.

Add reference(s) about the specificity/efficacy/toxicity of curcumin as inhibitor of aminopeptidase N.

The image quality of the cell images was very poor in the PDF files.

What is the impact of all the various peptides used in this work, of 0.05 % trypsin, of 50 microM curcumin, on HUVEC morphology/viability in standard 2D dishes?

Authors must provide more information about the HUVEC cells they used. Usually, HUVEC cells refer to primary cells, not to immortalized cell line, and they should be used at very low passage number (p1 to p6 max). In the manuscript it is mentioned that a HUVEC cell line was used (line 329) and for some experiments at passage 12 (line 412). Did they routinely perform mycoplasma tests?

For clarity, it would help to mention in the introduction that the YIGSR peptide was originally found in the laminin protein.

Reviewer #4 (Remarks to the Author):

The paper by Bitton and Matson reports an original concept on the activation of hidden signals in a synthetic extracellular matrix (ECM) inspired on natural cryptic sites. The concept is elegantly demonstrated through well-designed experiments and the use of appropriate analytic techniques. A significant amount of synthetic and characterization work was done to provide robust proof for the proposed concept. While the study is interesting, it requires a more convincing demonstration about the practical utility of the proposed concept on the activation of hidden signals in a synthetic ECM. In addition, the manuscript does not provide important information that is necessary for its comprehension and demonstration of significance. Some comments on these issues, are outlined below:

1) Background: Description of cryptic sites could be more detailed and supported with examples. For example, it is said that they "are short signaling peptides buried within the native extracellular matrix

(ECM)". However, it is not clear in what type of ECM. Collagen? Fibronectin or other glycoprotein? Please give more details.

2) Trypsin is used to start the cleavage of Lys and then promote the peptide switch, being essential to the process. This requires the supplementation of trypsin to start the cleavage, either in the solution experiments or incorporated in the gel. Later, a membrane enzyme was also tested. Figure 1A&B is confusing because in A it is shown a membrane enzyme activating the switch peptide, but in B the schematic shows the cleavage by trypsin, which is not a membrane enzyme. Is trypsin secreted by cells or is its supplementation required? Please clarify.

3) The concentration of trypsin is given in % (0.05%, 0.08%, 0.003%, 0.00025 wt.%) which is not conventional for reporting enzyme activity. It is typically given in U/mL, which gives the activity of the enzyme instead of concentration, being more relevant. The enzyme concentration/activity used was not the same in all experiments. Why? It would be useful to include the enzyme concentration/activity in the caption of Figure 1B as well.

4) Figure 3B shows a SEM image of the alginate gel, but it is not clear what is meant to be shown here. Some discussion should be included.

5) Was serum used in the cell culture experiments described in "Revealing cryptic sites in a synthetic ECM"? if yes, please comment on the use of serum in presence of trypsin.

6) The authors propose the application of a synthetic ECM with hidden signals in tissue engineering. However, this is not fully convincing because in such applications initial cell attachment is normally required for the survival of anchorage-dependent cells. It is not clear the advantage of having an inert matrix initially that then turns cell-adhesive. This is particularly relevant because cell-adhesive signals were used as hidden signals in this work. The appearance of other signals at later stages would be more sensible, but the authors should give examples of possible signals that would be beneficial of being presented a later stages. For example, the reverse case, starting with a cell-adhesive matrix and then turn into non adhesive would be useful for releasing cells.

Reviewer comments are provided below in *italics*, and our responses are shown below each comment. Manuscript text is shown in Arial font with specific sections added shown in red.

Reviewer #1

Specific Comment 1: Physical crosslinking of alginate biopolymers to yield hydrogel materials is performed with addition of CaCl₂. Can authors add few words about the limit of CaCl₂ addition, which would still produce sufficiently stiff viscoelastic materials (please add some data for storage and loss moduli) but at the same time not reduce enzymatic activity and mobility.

We thank the reviewer for bringing up this point, which we have now addressed in the manuscript, with the goal of not increasing the length too much. In brief, the CaCl₂ concentration was chosen based on literature precedent and our previous experience with alginate gels. In this study, the SEM image (Figure 3B) revealed pore sizes on the order of a few hundred nm, much larger than the size of the enzymes used here. We also included a frequency sweep rheological study of the hydrogel, with results now included in the extended data file (Fig S13). We did not do any further experiments to modify the CaCl₂ concentration or to determine its limit.

The manuscript now reads as follows (pg. 9):

Alginate hydrogels (physically crosslinked with Ca²⁺) are widely used as synthetic ECM materials. Their stiffness and porosity can be varied to allow proper mass transport and cell support. For hydrogels used in this study, enzymatic mobility is not likely to be significantly reduced, as can be expected from the large hydrogel pore size, which according to the SEM image (Figure 3B) is on the order of hundreds of nm, while still maintaining sufficient stiffness to provide cell support (fig. S13).

Fig. S13. Frequency sweep results of switch peptide-modified alginate hydrogel showing G' (circles) and G'' (squares).

Specific Comment 2: Looking at some of the advanced protected and labeled probes with in situ labelling of cellular integrins, it seems that a fair bit of molecular engineering was required. Can the authors comment on whether it was just solubility that was improved with added negative charged using the 5 glutamic acids in Ac-E5G2KSYIGRKR_B, or were they also needed to avoid unspecific binding on the cells or even cell uptake?

This is a good question, but there was not nearly as much molecular engineering as the reviewer may have perceived. We have worked extensively with amphiphilic peptides, so we knew that the rhodamine B (RB) unit would likely cause solubility problems. We added in the E5G2 sequence, forming Ac-E5G2KS_{YIG}RKR_B, simply to offset the hydrophobicity created by the RB unit. Because this worked well, we did not do any further modifications to arrive at an optimal sequence.

Specific Comment 3: Lipases can also be present in extracellular environments. At which point are O->N acyl shifts outcompeted with detrimental ester hydrolysis? Did authors observe any side reactions, and could a comment be added to the manuscript?

We thank the reviewer for the questions. Lipases or esterases certainly could lead to a loss of the ester functionality, although the results suggest that the steric bulk of the peptide makes any ester hydrolysis slow. In the basic switch peptide, ester hydrolysis would form the sequence KSR (or AKSR before APN-mediated cleavage). We cannot find any literature evidence that this trimer/tetramer binds integrins or has other specific bioactivity, so it would be similar in function to the original KS_{YIG}R peptide, i.e., it would show some non-specific binding, but no integrin-mediated cell adhesion. The fact that the switch peptide works in all the different contexts shown in this work suggests that ester hydrolysis, either by lipases/esterases or simply through non-enzymatic hydrolysis, must be substantially slower than the timescale of the switch reaction. We have added a comment on this to the manuscript.

The manuscript now reads as follows (pg. 15):

Quantitative cell imaging data (extended data fig. S16D) supported our qualitative conclusions, indicating that the AS_{YIG}RK-modified alginate hydrogel promoted cell spreading and adhesion of HUVECs through their own enzymatic activity, mimicking native cryptic sites in this proof-of-concept system. **We note that while esterases or lipases may slowly hydrolyze the depsipeptide ester bond, these cell studies suggest that any such reactions are substantially slower than the APN-mediated cleavage reaction because ester cleavage would result in a non-functional ASRK peptide.**

Reviewer #2

Specific Comment 1. The authors states “Activating Hidden Signals” in the title of the manuscript, but they don’t have any evidences to show any signal pathway had been activated in the manuscript, maybe the authors should re-consider the manuscript title.

The signal we were referring to in the manuscript title is hidden in the peptide itself, i.e., we are referring to the immobilized peptidic YIGSR. In short, YIGSR binds to integrins, which cluster in the plasma membrane, inducing recruitment and activation of intracellular signaling molecules. Our response to the next Specific Comment addresses this question of binding with additional experiments.

We have adjusted the manuscript text in the introduction to reflect this, and we thank the reviewer for pointing out that it was unclear in our original submission.

The manuscript now reads as follows (pg. 3):

We set out to synthesize a switch peptide that would rearrange into the bioadhesive YIGSR pentapeptide upon enzymatic activation (deprotection) by addition of trypsin and could be easily incorporated into a hydrogel. **We chose the YIGSR peptide, which is derived from the ECM protein laminin, because it binds to integrins, causing them to cluster in the plasma membrane and inducing recruitment and activation of intracellular signaling molecules.²⁶**

Additionally, we note that referring to peptide epitopes as signals is fairly common. For example, Stupp refers to “peptide-based supramolecular nanostructures that are intentionally designed to signal cells” (Peptide supramolecular materials for therapeutics, *Chem. Rev.*, **2018**, 47, 7539) and discusses a “high local concentration of signaling peptides on the PA fiber” (25th Anniversary Article: Supramolecular Materials for Regenerative Medicine, *Adv. Mater.*, **2014**, 1642). In a similar tone, Anseth mentions, in the context of peptide-containing materials, engineering of “bioactive scaffolds to signal to delivered cells or endogenous cells” (Thiol-ene and photo-cleavage chemistry for controlled presentation of biomolecules in hydrogels, *J. Cont. Release*, **2015**, 95). Guler discussed materials that “recapitulate the structure and function of the native extracellular matrix through signaling peptide epitopes” (Heparin mimetic peptide nanofiber gel promotes regeneration of full thickness burn injury, *Biomaterials*, **2017**, 134, 117). These are just a few examples that we quickly found in the recent literature. We therefore feel that the title accurately reflects the term ‘signal’ as it is used in the field of peptide-based biomaterials.

Specific Comment 2. The authors could consider to label some cell structure protein such as Focal Adhesion Kinase (FAK) or integrin when use the rhodamine B labeled peptide to treat the HUVEC to show the peptides are specific combined with cell structure and activate some signal pathway.

We thank the reviewer for the thoughtful suggestion, which we carried out as recommended. In short, we treated HUVECs with the rhodamine B labeled peptide Ac-YIGSR-K_{RB}, then labeled the cells with anti-CD29 (integrin β 1)-FITC. We observed that Ac-YIGSR-K_{RB} (red fluorescence) overlapped with anti-CD29-FITC labeled areas (green fluorescence), indicating that the laminin-derived peptide sequence YIGSR binds specifically to integrin β 1. We have included these data in the manuscript and extended data, and they now read as follows (pg. 9):

As a positive control, we treated cells with functional peptide Ac-YIGSR-K_{RB} for 30 h, which showed strong fluorescence at a level similar to that of switch peptide Ac-E₅G₂KS_{YIG}RK_{RB} treated with trypsin (extended data fig. S8 fourth row). Quantification of fluorescence micrographs for each treatment group (Fig. 2D) validated our conclusions. Finally, we also fixed the cells and labeled cell surface receptor CD29 (integrin 131, a laminin receptor), observing co-localization of the integrin stain with the rhodamine B-labeled peptide (fig. S9), confirming binding to integrin 131.

Fig. S9. Immunofluorescence staining of adhesion protein CD29 (integrin β 1, green) and rhodamine B labeled peptide Ac-YIGSR-K_{RB} (red) on HUVECs. HUVECs were treated with 50 μ M Ac-YIGSR-K_{RB} for 24 h, then cell nuclei were stained with 4,6-diamidino-2-phenylindole (DAPI, blue) and anti-CD29-FITC. Fluorescence microscopy showed that the laminin derived peptide Ac-YIGSR-K_{RB} colocalized with integrin 131 on the cell surface. Scale bar = 10 μ m.

Specific Comment 3. The authors should state in Fig 3, Fig S12 and Fig S14, which color represents living cells, which represents dead cells.

We thank the reviewer for catching this! We have made the changes in the manuscript and the SI.

The manuscript now reads as follows (page 11):

Fig 3. Revealing cryptic sites in a synthetic ECM leads to gain of biofunctionality. (A) Chemical structure of alginate–switch peptide hydrogel, gelled with Ca^{2+} . An amide bond links the -amine of the C-terminal Lys (blue) residue with the alginate backbone, while the -amine of the N-terminal Lys residue (pink) remains free. (B) SEM images of switch peptide-functionalized alginate hydrogel. Scalebar = 2 μm (200 nm for inset). (C) Schematic illustration describing gain of biofunctionality in the synthetic ECM activated by addition of trypsin. Trypsin cleaves the N-terminal Lys residue of the switch peptide, leading to rearrangement into a functional YIGSR sequence. (D) Representative brightfield, live/dead (green/red), and merged images of HUVEC cells after 72 h on unmodified alginate hydrogel, switch peptide-modified alginate hydrogel, functional peptide-modified alginate hydrogel, and switch peptide-modified alginate hydrogel with 0.003 wt.% trypsin added. Scale bar represents 200 μm . (E) Schematic illustration describing gain of biofunctionality in the synthetic ECM activated by cell surface APN enzymes. APN cleaves the N-terminal Ala residue of the switch peptide, leading to rearrangement into a functional YIGSR sequence. (F) Representative brightfield, live/dead (green/red), and merged images of HUVEC cells after 72 h on ASYIGRK-modified alginate hydrogel, ASYIGRK-modified alginate hydrogel with curcumin (inhibitor, 50 M), KSYIGRK switch peptide-modified alginate hydrogel, and functional peptide-modified alginate hydrogel. Scale bar represents 100 μm .

The extended data now reads as follows (page S17, S20):

Fig. S14. Representative brightfield, live/dead (green/red), and merged images of HUVEC cells adhered to unmodified alginate hydrogel (alginate), switch peptide-modified alginate hydrogel (KS_{YIG}RK-Alg), switch peptide-modified alginate hydrogel with 0.05% trypsin added (final concentration 0.003%, KS_{YIG}RK-Alg + trypsin), and functional peptide-modified alginate hydrogel (YIGSRK-Alg) for **(A)** 24 h post-seeding. Live/dead staining and subsequent fluorescence microscopy were performed to confirm cell viability and cell spreading. Scale bars represent 200 μm . Magnification = 10 \times . **(B)** 24 h post-seeding. Live/dead staining and subsequent fluorescence microscopy was performed to confirm cell viability and cell spreading. Scale bars represents 200 μm . Magnification = 4 \times . **(C)** 72 h post-seeding. Live/dead staining and subsequent fluorescence microscopy was performed to confirm cell viability and cell spreading. Scale bars represents 200 μm . Magnification = 4 \times . See 72 h post-seeding, magnification = 10 \times images in **Fig 3D**. **(D)**. Corrected total cell fluorescence (CTCF) values measured from fluorescence images of peptide-alginate hydrogels mentioned above at 24 h and 72 h. Average fluorescence intensities were quantified by ImageJ (cell counts are >30 for each group from three separate wells).

Fig. S16. Representative brightfield, live/dead (green/red), and merged images of HUVEC cells adhered to unmodified alginate hydrogel, AS_{YIG}RK-modified alginate hydrogel, AS_{YIG}RK-modified alginate hydrogel with curcumin (50 μM) added, switch peptide-modified alginate hydrogel, functional peptide-modified alginate hydrogel, and

YIGSRK-modified alginate hydrogel with curcumin (50 μ M) added for **(A)** 24 h post-seeding. Live/dead staining and subsequent fluorescence microscopy were performed to confirm cell viability and cell spreading. Scale bar represents 100 μ m. Magnification = 10 \times . **(B)** 24 h post-seeding. Live/dead staining and subsequent fluorescence microscopy were performed to confirm cell viability and cell spreading. Scale bar represents 200 μ m. Magnification = 4 \times . **(C)** 72 h post-seeding. Live/dead staining and subsequent fluorescence microscopy were performed to confirm cell viability and cell spreading. Scale bar represents 200 μ m. Magnification = 4 \times . See 72 h post-seeding, magnification = 10 \times images in **Fig 3F**. **(D)** Corrected total cell fluorescence (CTCF) values measured from fluorescence images of peptide-alginate hydrogels mentioned above at 24 h and 72 h. Average fluorescence intensities were quantified by ImageJ (cell counts are >30 for each group from three separate wells).

Specific Comment 4. The authors should make statistics test to check if there are significant differences in fig 2D, fig S8B, fig S12D and fig S14D.

We thank the reviewer for this suggestion, and we agree that it is useful to include this as well. We have now run a one-way analysis of variance (ANOVA) with Student–Newman–Keuls comparisons post hoc tests on the suggested figures. We include the results as separate tables in the extended data section (tables S1, S2, S3) The updated figures with statistics tests and the extended data now read as follows (page S12, S17, S20):

Page S12

Table S1. Statistics Data of CTCF at each timepoint measured from fluorescence images of Ac-E5G2KS_{YIGR}-K_{RB} with trypsin, and control peptides mentioned above at 30 h.

Comparison	Mean Difference	q	P value
1 h vs 8 h	-80919	2.708	ns
8 h vs 24 h	-141239	6.524	***
24 h vs 30 h	-134325	5.454	***
30 h vs Ac-YIGSR-K _{RB}	-89248	3.355	*
30 h vs Ac-E5G2KS _{YIGR} -K _{RB}	-275564	16.54	***
30 h vs Ac-K _{YIGR} -K _{RB}	-358396	18.03	***

30 h vs Ac-K _{YIG} R-K _{RB} + trypsin	-356482	14.426	***
Ac-YIGSR-K _{RB} vs Ac-E ₅ G ₂ K _{SYIG} R-K _{RB}	-447648	20.987	***
Ac-YIGSR-K _{RB} vs Ac-K _{YIG} R-K _{RB}	-445730	17.214	***
Ac-YIGSR-K _{RB} vs Ac-K _{YIG} R-K _{RB} + trypsin	-447644	20.986	***
Ac-E ₅ G ₂ K _{SYIG} R-K _{RB} vs Ac-K _{YIG} R-K _{RB}	-3.909	na	ns
Ac-E ₅ G ₂ K _{SYIG} R-K _{RB} vs Ac-K _{YIG} R-K _{RB} + trypsin	-3.764	na	ns
Ac-K _{YIG} R-K _{RB} vs Ac-K _{YIG} R-K _{RB} + trypsin	-0.145	na	ns

Group comparisons are indicated as determined by a one-way analysis of variance (ANOVA) with a Student–Newman–Keuls comparisons post hoc test. *** indicates $p < 0.001$, ** indicates $p < 0.01$, * indicates $p < 0.05$, and *ns* indicates no significance among indicated treatment groups.

Page S17:

Table S2. Statistics Data of CTCF measured from fluorescence images of peptide-alginate hydrogels mentioned above at 24 h and 72 h.

Comparison	Mean Difference	q	P value
24 h K _{SYIG} RK-Alg vs 24 h K _{SYIG} RK-Alg+trypsin	-75122	6.679	***
24 h K _{SYIG} RK-Alg vs 24 h YIGSRK-Alg	-184904	17.217	***
24 h K _{SYIG} RK-Alg+trypsin vs 24 h YIGSRK-Alg	-109782	10.222	***
24 h K _{SYIG} RK-Alg+trypsin vs 72 h K _{SYIG} RK-Alg+trypsin	-92959	8.536	***
24 h K _{SYIG} RK-Alg vs 72 h K _{SYIG} RK-Alg	-6657.8	na	ns

72 h KS _{YIG} RK-Alg vs 72 h KS _{YIG} RK-Alg+trypsin	-161423	15.987	***
72 h KS _{YIG} RK-Alg vs 72 h YIGSRK-Alg	-213882	22.436	***
72 h KS _{YIG} RK-Alg+trypsin vs 72 h YIGSRK-Alg	-52459	5.256	***
24 h YIGSRK-Alg vs 72 h YIGSRK-Alg	-35636	3.63	*
72 h KS _{YIG} RK-Alg+trypsin vs 24 h YIGSRK-Alg	-16823	1.623	ns

Group comparisons are indicated as determined by a one-way analysis of variance (ANOVA) with a Student–Newman–Keuls comparisons post hoc test. *** indicates $p < 0.001$, ** indicates $p < 0.01$, * indicates $p < 0.05$, and *ns* indicates no significance among indicated treatment groups.

Page S20:

Table S3. Statistics Data of CTCF measured from fluorescence images of peptide-alginate hydrogels mentioned above at 24 h and 72 h.

Comparison	Mean Difference	q	P value
24 h AS _{YIG} RK-Alg vs 72 h AS _{YIG} RK-Alg	-49904	4.497	**
24 h AS _{YIG} RK-Alg vs 24 h AS _{YIG} RK-Alg + cur.	-172006	13.662	***
24 h AS _{YIG} RK-Alg vs 24 h KS _{YIG} RK-Alg	-117751	10.433	***
24 h AS _{YIG} RK-Alg vs 24 h YIGSRK-Alg	-58276	5.656	***
24 h AS _{YIG} RK-Alg + cur. vs 24 h KS _{YIG} RK-Alg	-4349.9	na	ns
24 h AS _{YIG} RK-Alg + cur. vs 24 h YIGSRK-Alg	-180377	15.162	***
24 h AS _{YIG} RK-Alg + cur. vs 72 h AS _{YIG} RK-Alg + cur.	-682.01	na	ns

24 h KS _{YIG} RK-Alg vs 24 h YIGSRK-Alg	-176027	17.085	***
24 h KS _{YIG} RK-Alg vs 72 h KS _{YIG} RK-Alg	-401.37	na	ns
24 h YIGSRK-Alg vs 72 h YIGSRK-Alg	-34586	3.596	*
72 h AS _{YIG} RK-Alg vs 72 h AS _{YIG} RK-Alg + cur.	-172688	14.166	***
72 h AS _{YIG} RK-Alg vs 72 h KS _{YIG} RK-Alg	-167254	15.339	***
72 h AS _{YIG} RK-Alg vs 72 h YIGSRK-Alg	-47610	4.561	**
72 h AS _{YIG} RK-Alg + cur. vs 72 h KS _{YIG} RK-Alg	-5433.3	na	ns
72 h AS _{YIG} RK-Alg + cur. vs 72 h YIGSRK-Alg	-220298	18.705	***
72 h KS _{YIG} RK-Alg vs 72 h YIGSRK-Alg	-214865	20.582	***
72 h AS _{YIG} RK-Alg vs 24 h YIGSRK-Alg	-8371.6	0.8293	ns
24 h AS _{YIG} RK-Alg vs 24 h YIGSRK-Alg + cur.	-39800	3.940	**
24 h YIGSRK-Alg + cur. vs 72 h YIGSRK-Alg + cur.	-16805	na	ns
24 h YIGSRK-Alg + cur. vs 24 h YIGSRK-Alg	-23128	2.615	ns
72 h AS _{YIG} RK-Alg vs 72 h YIGSRK-Alg + cur.	-6700.2	na	ns
72 h YIGSRK-Alg + cur. vs 72 h YIGSRK-Alg	-40910	4.373	**

Group comparisons are indicated as determined by a one-way analysis of variance (ANOVA) with a Student–Newman–Keuls comparisons post hoc test. *** indicates $p < 0.001$, ** indicates $p < 0.01$, * indicates $p < 0.05$, and ns indicates no significance among indicated treatment groups.

Specific Comment 5. *The authors should improve the image's quality. It is difficult to identify the cells in the Fig 3D row 2 brightfield image, Fig 2D row 2 brightfield image, while there are strong fluorescence signals in the relative fluorescence images.*

We thank the reviewer for pointing this out. We have adjusted brightness in the suggested images to improve their quality. We hope that including the additional figure files in this submission will also help in the event that the pdf conversion is the cause of the poor image quality.

Specific Comment 6. *The manuscript writing is a little confusion. The extended fig S3, fig S7 and fig S10 are not mentioned in the main text. The fig S6 is mentioned in line 102, while the fig S5 is mentioned in line 137 and fig S4 is mentioned in line 251, it is difficult to read the results.*

We thank the reviewer for pointing out this concern. We now refer to figures S1-S4, which show chemical structures and mass spectra, early in the manuscript. We also switched the order of fig S5 and fig S7 (previously S6, MALDI-TOF data) to avoid the confusing order noted above. We agree with the reviewer that fig S6 (previously S7, cell viability data) and fig S11 (previously S10, chemical structure of alginate after peptide addition) should be explicitly mentioned in the manuscript.

The manuscript now reads as follows (page 4, page 6 and page 12):

Page 4: As the first step toward mimicking cryptic site function, we designed and synthesized (extended data fig. S1-4, **scheme S1-3**) a non-functional depsipeptide that could be enzymatically deprotected to reveal a functional peptide epitope.

Page 6: The design of this peptide included the E₅G₂ sequence on the N-terminus to increase peptide solubility, counteracting the hydrophobic K_{RB} unit that was added to the C-terminus. **Cell viability studies (extended data fig. S6) showed no cytotoxicity up to 200 μM for peptides KS_{YIG}R, YIGSR, or any RB labeled peptides.**

Page 12: To conjugate the switch peptide to an alginate backbone, we utilized carbodiimide chemistry, in which an amine group from the peptide reacts with alginate carboxyl groups to form an amide bond, a method widely used in alginate modification (**extended data fig. S11**).

Reviewer #3

Specific Comment 1. I have one very major concern that should be experimentally addressed. Curcumin was shown to be very toxic on many cell types, 50 microM is a very high concentration. The important results of Fig. 3F, fourth row, may be explained by the toxicity of the curcumin, instead of by its inhibition of aminopeptidase N (APN) activity. An important experiment is lacking: the adhesion of HUVEC to YIGSRK-Alg in presence of 50 microM curcumin. If curcumin toxicity may turn out to be a problem, the authors could try to knock-down by siRNA the CD13 gene expressing APN, or to compare cell lines with different expression levels/activities of APN.

This is a very good point, and we overlooked the potential toxicity of curcumin in our original submission. We thank the reviewer for pointing this out. We have now done the suggested experiment (YIGSRK-Alg hydrogel in presence of 50 μ M curcumin) as well as a cell viability assay in 2D of 50 μ M curcumin. We also note below several papers that report high viability of HUVECs (>80%) treated with curcumin (up to 100 μ M) for up to 72 h:

- Kumar, A., Dhawan, S., Hardegen, N. J. & Aggarwal, B. B. Curcumin (Diferuloylmethane) Inhibition of Tumor Necrosis Factor (TNF)-Mediated Adhesion of Monocytes to Endothelial Cells by Suppression of Cell Surface Expression of Adhesion Molecules and of Nuclear Factor- κ B Activation. *Biochem. Pharmacol.* **55**, 775–783 (1998).
- Do, X.-H. *et al.* Differential Cytotoxicity of Curcumin-Loaded Micelles on Human Tumor and Stromal Cells. *International Journal of Molecular Sciences* vol. 23 (2022).
- Ouyang, J., Li, R., Shi, H. & Zhong, J. Curcumin Protects Human Umbilical Vein Endothelial Cells against H₂O₂-Induced Cell Injury. *Pain Res. Manag.* **2019**, 3173149 (2019).
- Hosseini, A. *et al.* Curcumin modulates the angiogenic potential of human endothelial cells via FAK/P-38 MAPK signaling pathway. *Gene* **688**, 7–12 (2019).
- Yuan, Y. *et al.* Curcumin improves the function of umbilical vein endothelial cells by inhibiting H₂O₂-induced pyroptosis. *Mol Med Rep* **25**, 214 (2022).
- We also noticed below some studies of relatively high toxicity of curcumin on HUVECs:
- Zhao, L. *et al.* Curcumin protects human umbilical vein endothelial cells against high oxidized low density lipoprotein-induced lipotoxicity and modulates autophagy. *Iran. J. Basic Med. Sci.* **24**, 1734–1742 (2021).

In our experiments, we observed 93% cell viability in HUVECs treated with 50 μ M curcumin after 24 h treatment, and 85% cell viability after 72 h. These results are now included in the extended data section (fig. S6C). The YIGSRK-modified alginate hydrogel with 50 μ M curcumin also showed a similar number of live cells adhered to the hydrogel after 24 h or 72 h treatment compared to the YIGSRK-modified alginate hydrogel alone. These results are now included in the extended data section (fig. S16). The extended data section now reads as follows (page S9, page S19):

Fig. S6. Cell viability of HUVECs treated with **(A)** different concentrations of KS_{Y16}R, Ac-YIGSR, AS_{Y16}RK, KS_{Y16}RK, and Ac-YIGSRK for 24 h; **(B)** different concentrations of rhodamine B-labeled peptides Ac-E₅G₂-KS_{Y16}R-K_{RB}, Ac-YIGSR-K_{RB}, and Ac-K_{Y16}R-K_{RB}. **(C)** 50 μ M curcumin and 0.003% trypsin for 24 h and 72 h.

Fig. S16. Representative brightfield, live/dead (green/red), and merged images of HUVEC cells adhered to unmodified alginate hydrogel, AS_{YIG}RK-modified alginate hydrogel, AS_{YIG}RK-modified alginate hydrogel with curcumin (50 μM) added, switch peptide-modified alginate hydrogel, functional peptide-modified alginate hydrogel, and

YIGSRK-modified alginate hydrogel with curcumin (50 μ M) added for (A) 24 h post-seeding. Live/dead staining and subsequent fluorescence microscopy were performed to confirm cell viability and cell spreading. Scale bar represents 100 μ m. Magnification = 10 \times . (B) 24 h post-seeding. Live/dead staining and subsequent fluorescence microscopy were performed to confirm cell viability and cell spreading. Scale bar represents 200 μ m. Magnification = 4 \times . (C) 72 h post-seeding. Live/dead staining and subsequent fluorescence microscopy were performed to confirm cell viability and cell spreading. Scale bar represents 200 μ m. Magnification = 4 \times . See 72 h post-seeding, magnification = 10 \times images in Fig. 3F. (D) Corrected total cell fluorescence (CTCF) values measured from fluorescence images of peptide-alginate hydrogels mentioned above at 24 h and 72 h. Average fluorescence intensities were quantified by ImageJ (cell counts are >30 for each group from three separate wells).

Specific Comment 2. I strongly recommend to move to main figures some data that are now presented in extended data section, since they are crucial information. Move Fig. S8 to Fig. 2. Move quantification of Fig. S12D and Fig. S14D to Fig. 3.

We thank the reviewer for the suggestion.

Figure S8 contains two components: A) Control studies relevant to Figure 2, and B) a more comprehensive version of the corrected total cell fluorescence (CTCF) data in Figure 2B. The control studies show three rows in fig S8A with no fluorescence, as expected. The fourth row in fig S8A shows a positive control. While we feel that the more comprehensive timecourse study images and the quantified CTCF data are not needed in the main manuscript, we agree with the reviewer that the 4th row in fig S8a could be useful in Figure 2. Therefore, we have added this to the manuscript. The updated version of Figure 2 appears below:

Fig 2. Switch peptide rearrangement leads to gain of functionality. (A) Schematic illustration of HUVEC cells in culture with fluorescently labeled switch peptide ($E_5G_2KS_{YIG}R-K_{RB}$). Upon activation by trypsin, the peptide rearranges into a fluorescently labeled functional peptide, $YIGSRK_{RB}$, where the $YIGSR$ epitope binds to cell surface integrins. Peptide $E_5G_2KS_{YIG}R-K_{RB}$ is also fluorescent, but it cannot bind to integrins and is removed in the washing step. (B) Chemical structure of fluorescently labeled switch peptide. (C) Brightfield, fluorescence, and merged images of HUVECs preincubated with $Ac-E_5G_2KS_{YIG}R-K_{RB}$ and trypsin at various timepoints, and positive control $Ac-YIGSR-K_{RB}$ at 30 h. Scale bar = 100 μm . (D) Corrected total cell fluorescence (CTCF) values at each timepoint measured from fluorescence images. Statistical tests are included in the extended data (table S1).

As for the quantification in fig S12d and fig S14d, we do not feel that inclusion of these graphs into Fig. 3 would improve the manuscript because the images already tell the story clearly. The differences in the live cell (calcein AM) fluorescence for the various treatment groups are quite stark. We feel that squeezing the cell images down to accommodate the bar graphs in fig S12d and S14d in a figure that is already full-width and takes up most of the page vertically would not add to the story. While we agree with the reviewer that these graphs are crucial information, it is the same information that is already conveyed in the cell images.

Specific Comment 3. How many times were reproduced the experiments presented in Fig.2 and Fig.3?

In the original manuscript, we noted the following in the cell viabilities assay subsection of the Methods section (pg 19):

Mean values are reported together with the standard error of the mean (SEM) representing the combination of 3 independent experimental runs with five replicates per experiment.

Because we aimed to keep the captions as short as possible, we decided to just mention this once in the Methods section.

Specific Comment 4. Please, explain better the CTCF quantifier.

We thank the reviewer for pointing out that our method for cell total calculated fluorescence (CTCF) was not fully described. We have now included a discussion of this method in the extended data file.

The explanation now reads as follows (extended data file page S11):

The corrected total cell fluorescence (CTCF) was calculated as follows^{1,2} (Ref. 1-2 in extended data file):

CTCF = integrated density – (area of selected cell × mean fluorescence of background readings)

Specific Comment 5. Add reference(s) about the specificity/efficacy/toxicity of curcumin as inhibitor of aminopeptidase N.

We thank the reviewer for suggesting that we reference curcumin as an inhibitor of APN, and we apologize for overlooking this.

The manuscript now reads as follows (page 14):

Ala-MCA showed negligible fluorescence itself, but in the presence of HUVECs, strongly fluorescent 7-amino-4-methylcoumarin (AMC) was released, confirming APN activity. We also showed that **APN in HUVECs could be inhibited using 50 μ M curcumin, based on the reported IC50 value of 10 μ M,³⁶ (Ref. 36 in main) substantially decreasing fluorescence intensity (extended data fig. S15). We also found that curcumin was not toxic to HUVECs at this level (extended data fig S7C, S16).**

Specific Comment 6. *The image quality of the cell images was very poor in the PDF files.*

We apologize for the poor image quality. We prepared high resolution images in Illustrator, and they looked okay in the pdf proof, but it seems that they degraded further during processing. We have included additional image files here in hopes that their quality is improved.

Specific Comment 7. *What is the impact of all the various peptides used in this work, of 0.05 % trypsin, of 50 microM curcumin, on HUVEC morphology/viability in standard 2D dishes?*

We thank the reviewer for the question, and we agree these results are indeed important in all of the *in vitro* experiments of this work. We now have conducted and included cell viability assay for all the peptides we used in this work (up to 200 μ M), as well as 0.05% trypsin (maximum final concentration was 0.003%) and 50 μ M curcumin on HUVECs in standard 2D experiments. The results indicate that HUVECs retain high viability (>90% in nearly all cases) for all of the treatments. The manuscript and extended data now read as follows (page 6, page 12, page S9):

Page 6:

The design of this peptide included the E₅G₂ sequence on the N-terminus to increase peptide solubility, counteracting the hydrophobic K_{RB} unit that was added to the C-terminus. **Cell viability studies (extended data fig. S6) showed no cytotoxicity up to 200 μ M for peptides KS_{YIG}R, YIGSR, or any RB-labeled peptides.**

Page 12:

We also prepared an authentic functional peptide Ac-YIGSRK for control studies. **Cell viability studies (extended data fig. S6) showed no cytotoxicity up to 200 μ M for peptides KS_{YIG}RK, Ac-YIGSRK, KS_{YIG}RK, trypsin, and 50 μ M curcumin.**

Fig. S6. Cell viability of HUVECs treated with (A) different concentrations of KS_{Y1G}R, Ac-YIGSR, AS_{Y1G}RK, KS_{Y1G}RK, and Ac-YIGSRK for 24 h; (B) different concentrations of rhodamine B labeled peptides Ac-E₅G₂-KS_{Y1G}R-K_{RB}, Ac-YIGSR-K_{RB}, and Ac-K_{Y1G}R-K_{RB}. (C) 50 μM curcumin and 0.003% trypsin for 24 h and 72 h.

Specific Comment 8. Authors must provide more information about the HUVEC cells they used. Usually, HUVEC cells refer to primary cells, not to immortalized cell line, and they should be used at very low passage number (p1 to p6 max). In the manuscript it is mentioned that a HUVEC cell line was used (line 329) and for some experiments at passage 12 (line 412). Did they routinely perform mycoplasma tests?

The HUVEC cells were purchased from ATCC (Manassas, VA, USA). We apologize for suggesting that they are a cell line, which is not the case—they are primary cells as the reviewer notes. Regarding passage number, we followed recommendations from a 2014 paper on culture of HUVECs (Chen, “Effects of long-term serial cell passaging on cell spreading, migration, and cell-surface ultrastructures of cultured vascular endothelial cells” *Cytotechnology*, **2014**, 66, 229-238), in which they studied up to passage 35. The authors of this paper note that “Percentage-based spreading assay showed that the average percentages of spread cells before 10 passages were similar whereas the values gradually decreased after the 15th passage” and they further recommend that “HUVEC cells at a passage of less than 10 are optimum for studies.” Nearly all of our studies were conducted in this passage range, with just the APN studies conducted at later passage numbers (up to 12). Because drop-offs in cell activity did not become prominent in the Chen study until passage 15, we opted to continue with the same cells out to passage 12 rather than start a new vial (Chen and coworkers noted some changes in passages 1-5). We have adjusted the text to reflect the range of passage numbers used for the various cell studies in the Cell Culture section.

The manuscript now reads as follows (pg. 18):

Cell studies were conducted using human umbilical vein endothelial cells (HUVECs) ~~cell line~~ at passages 5–12.

Regarding mycoplasma, we have performed mycoplasma tests on these HUVEC cells, and we found them to be negative every time.

Specific Comment 9. For clarity, it would help to mention in the introduction that the YIGSR peptide was originally found in the laminin protein.

This is a good point, and we thank the reviewer for suggesting it. While we mentioned laminin in the results and discussion section, we neglected to mention it in the introduction. We have now modified the introduction to mention that YIGSR was originally found in laminin.

The manuscript now reads as follows (page 3):

We set out to synthesize a switch peptide that would rearrange into the bioadhesive YIGSR pentapeptide upon enzymatic activation (deprotection) by addition of trypsin and could be easily incorporated into a hydrogel. ~~We chose the YIGSR peptide, which is derived from the ECM protein laminin, because it binds to integrins, causing them to cluster in the plasma membrane and inducing recruitment and activation of intracellular signaling molecules.~~²⁶

Reviewer #4

Specific Comment 1. *Background: Description of cryptic sites could be more detailed and supported with examples. For example, it is said that they "are short signaling peptides buried within the native extracellular matrix (ECM)". However, it is not clear in what type of ECM. Collagen? Fibronectin or other glycoprotein? Please give more details.*

Most cryptic sites have been found in collagen, but they are by no means limited to collagen. In previous versions of the manuscript, we had a longer discussion of native cryptic sites, but we cut this down to keep within the typical introduction length for *Nature Communications*. We have added back in an additional example sentence to the introductory paragraph. We agree with the reviewer that a little additional background discussion will improve the manuscript.

The manuscript now reads as follows (page 2):

The ECM, a network of proteins and polysaccharides that structurally supports cells and organs,² contains buried peptidic signals called cryptic sites,¹ which are inaccessible to cells until enzymatic degradation^{3,5} or mechanical stress⁴ triggers a structural alteration to uncover these short peptides.^{4,6-8} Once revealed, cryptic site peptide sequences become available to bind specific cellular receptors, providing instructions to cells to initiate behavioral changes, making them a vital component in the cell–ECM synergy.^{1,9-12} **For example, proteolytic cleavage of collagen IV by MMP-9 exposes a cryptic site hidden within its triple helical structure that is required for angiogenesis.⁵** Synthetic hydrogels offer the possibility of mimicking the structural features of native ECM, with vast potential biomedical applications.¹³⁻¹⁷

Specific Comment 2. *Trypsin is used to start the cleavage of Lys and then promote the peptide switch, being essential to the process. This requires the supplementation of trypsin to start the cleavage, either in the solution experiments or incorporated in the gel. Later, a membrane enzyme was also tested. Figure 1A&B is confusing because in A it is shown a membrane enzyme activating the switch peptide, but in B the schematic shows the cleavage by trypsin, which is not a membrane enzyme. Is trypsin secreted by cells or is its supplementation required? Please clarify.*

We thank the reviewer for pointing out potential confusion based on Figure 1A, which is a cartoon image showing the overall concept for the paper and includes the membrane enzyme APN as the trigger, and Figure 1B, which shows the chemical structure of the switch peptide that is triggered by external addition of trypsin. If these were separate figures instead of panels A and B within the same figure, the distinction might be more clear, but figures with several panels are typical of *Nature Communications*. To specifically answer the question, trypsin must be added to trigger the switch for the peptide shown in Figure 1B. To keep within the typical style of this journal, we have not adjusted the figure, but instead we have added details to the caption to improve clarity.

Figure 1 (not changed) and its caption (updated) are shown below:

Fig. 1. Design of cryptic sites in a synthetic ECM using switch peptides. (A) Schematic illustration of synthetic ECM with switch peptides as cryptic site mimics. Cell surface enzymes remove an amino acid residue, activating the switch peptide and forming the cell-adhesive YIGSR sequence. **(B)** Chemical transformation of the switch peptide $KS_{YIG}R$ containing a "split sequence" where the Ser residue is attached to a side chain YIG sequence (red) through its side chain alcohol, forming an ester bond. Removal of the N-terminal Lys residue by added trypsin reveals the free Ser amine (intermediate peptide). Once the free amine is present, a spontaneous O \rightarrow N acyl shift occurs, generating the native peptide bond and forming the functional YIGSR peptide (blue). **Other examples in this report use a membrane enzyme to trigger the switch.** **(C)** MALDI-TOF spectra of switch peptide after incubation with trypsin for different time points. Peak $m/z = 764$ ($[M+H]^+$) indicates the switch peptide, while $m/z = 636$ ($[M+H]^+$) indicates the functional peptide.

Specific Comment 3. The concentration of trypsin is given in % (0.05%, 0.08%, 0.003%, 0.00025 wt.%) which is not conventional for reporting enzyme activity. It is typically given in U/mL, which gives the activity of the enzyme instead of concentration, being more relevant. The enzyme concentration/activity used was not the same in all experiments. Why? It would be useful to include the enzyme concentration/activity in the caption of Figure 1B as well.

We agree that enzymes are typically reported in U/mL. However, we reported trypsin in wt% because this is how it is labeled by the supplier. Since it is typically used in cell culture to remove cells from the culture dish in a range of 0.05-0.25 wt%, we used the wt% units

throughout so that readers familiar with cell culture would quickly understand that the amounts used were far less than is typically needed for cleavage from a cell culture dish. Different concentrations were used for the different assays (MALDI-TOF, cell culture) for various experimental reasons, but all were within a similar range (we note that 0.05 wt% was the stock solution used in the cell release steps in cell culture, but triggering of the switch required much less, ranging from 0.00025 to 0.003 wt%).

***Specific Comment 4.** Figure 3B shows a SEM image of the alginate gel, but it is not clear what is meant to be shown here. Some discussion should be included.*

We thank the reviewer for noting that we should add some short discussion on the SEM image. We have now done so, and we refer the reviewer to our response to Reviewer 1, Specific Comment 1.

***Specific Comment 5.** Was serum used in the cell culture experiments described in "Revealing cryptic sites in a synthetic ECM"? if yes, please comment on the use of serum in presence of trypsin.*

We thank the reviewer for the questions and we apologize for the vague description. For the cell culture experiments in "Revealing cryptic sites in a synthetic ECM" session, we used serum free endothelial cell growth medium (ECGM) to avoid any variables from fetal bovine serum (FBS), which might contain protease inhibitors. For the "Gain of biological function" session, we first used 10% FBS containing ECGM for cell culture, then we washed the cells three times with PBS and used serum free ECGM for the further treatments and fluorescence study. In summary, we used serum-free ECGM in the presence of trypsin treatment in cell culture experiments. We have added some descriptions in the experimental details.

The manuscript now reads as follows (page 21):

Page 20: HUVEC cells were seeded on the top of the hydrogels at a density of 10,000 **cells per gel with 150 μ L serum-free** ECGM media. Cells were cultured in an incubator for 24 h or 72 h. **After each incubation time, the media was removed and 150 μ L fresh serum-free** ECGM media was added to each well.

***Specific Comment 6.** The authors propose the application of a synthetic ECM with hidden signals in tissue engineering. However, this is not fully convincing because in such applications initial cell attachment is normally required for the survival of anchorage-dependent cells. It is not clear the advantage of having a inert matrix initially that then turns cell-adhesive. This is particularly relevant because cell-adhesive signals were used as hidden signals in this work. The appearance of other signals at later stages would be more sensible, but the authors should give examples of possible signals that would be beneficial of being presented a later stages. For example, the reverse case, starting with a cell-adhesive matrix and then turn into non adhesive would be useful for releasing cells.*

This is a good point, and we thank the reviewer for bringing it up. The goal here was to use cell adhesion as a proof of principle because it is easy to visualize this gain of biofunctionality. We agree that the appearance of other signals at later stages, for example in stem cell differentiation, would likely be more useful. The reviewer's suggestion to release cells via a turn-on mechanism is also useful. While this was mentioned in the original conclusion section, we have added some additional language on this topic to make the utility of this approach more convincing.

The manuscript now reads as follows (page 16):

These findings show that through chemical design, a synthetic ECM can gain biofunctionality upon enzymatic action. Potential applications include tissue engineering and regenerative medicine approaches that enable cells and their synthetic ECM environment to change over time, for example during stem cell differentiation processes, **where specific signals are often required at certain times in the differentiation process**. Moving forward, the switch peptide approach may enable various depsipeptides to rearrange into a wide variety of functional epitopes; other specific stimuli in addition to enzymes can also be envisioned including changes in pH or the presence of certain metabolites capable of cleaving a specific bond. Elevated enzyme activities and high levels of reactive metabolites are often present in aberrant microenvironments; thus, these tools and approaches provide the potential for precise theranostic biomaterial designs that mimic critical dynamic aspects of native ECM.

Finally, we note that this methodology could also be used to select for certain cell types that can activate the hidden signals and bind the matrix. In other words, the capacity for specific activation could be widely applicable.

REVIEWERS' COMMENTS

Reviewer #2 (Remarks to the Author):

The authors had done a lot of experiments and analyzes to improve the manuscript's quality, they had addressed most of my concerns. But it is a pity that some figure's quality and detail still need to be improved.

1. In Fig. 2C 8h brightfield image, it has some wrinkles, it is difficult to recognize the cells' boundaries.
2. In Fig.3F YIGSRK-Alg and ASYIGRK-Alg brightfield images, the backgrounds are too complex, it is very easy to make the reader confused with the cells with the matrix.
3. There are too much bubbles in Fig. 3D KSYIGRK-Alg + trysin brightfiels image, some bubbles are co-localization with the cell Live/Dead fluorescence, some bubbles had not Live/Dead fluorescence. It is difficult to recognize which are the cells, which are the bubbles.

The authors stated they had 3 independent experimental runs with five replicates per experiment in Fig. 2 and Fig. 3, I think they can choose some better representative images.

4. In Fig. S6A, the authors used 3 kinds of gray bars to present for different groups, I think they can use different colors to present different groups. In Fig. S6B, they used two kinds of red colors to present Ac-E5G2-KSYIGR-KRB and Ac-KYIGR-KRB, I think they can use black color to instead of one kind of red color.

Reviewer #3 (Remarks to the Author):

The authors answer in a satisfactory way to all my questions. In particular, they performed and they now report in the manuscript the control experiment assessing the adhesion of HUVEC to YIGSRK-Alg in presence of 50 microM curcumin. I recommend the acceptance of this original and interesting work.

Reviewer #4 (Remarks to the Author):

My questions have been clarified and no further clarifications/revisions are requested.

Reviewer comments are provided below in italics, and our responses are shown below each comment. Updated figures are included in the manuscript file.

Reviewer #2

Specific Comment 1. In Fig. 2C 8h brightfield image, it has some wrinkles, it is difficult to recognize the cells' boundaries.

We thank the reviewer for this comment. We have replaced it with a better image.

Specific Comment 2. In Fig. 3F YIGSRK-Alg and ASYIGRK-Alg brightfield images, the backgrounds are too complex, it is very easy to make the reader confused with the cells with the matrix.

The reviewer notes that the mixture of cells on top of matrix make it hard to make out the cells in the brightfield images. However, this highlights that the cells are interacting with the matrix. Also, the cells are visible in the fluorescence image, so this should eliminate any confusion in the 'Merge' column. We searched all of our images for the studies shown in Figure 3F, but we could not find any in which the cells were easier to see in the brightfield image. We feel that this image accurately reflects what was observed.

Specific Comment 3. There are too much bubbles in Fig. 3D KSYIGRK-Alg + trysin brightfields image, some bubbles are co-localization with the cell Live/Dead fluorescence, some bubbles had not Live/Dead fluorescence. It is difficult to recognize which are the cells, which are the bubbles. The authors stated they had 3 independent experimental runs with five replicates per experiment in Fig. 2 and Fig. 3, I think they can choose some better representative images.

This is a good point. The bubbles form during hydrogel preparation, and some tend to get stuck in the gel. We searched all of the images obtained for the experiment shown in Figure 3D, and we have replaced the original image with one that has fewer bubbles. Brightfield images of gels are typically less well defined than those on 2D plates because the tops of the gels are not perfectly flat.

Specific Comment 4. In Fig. S6A, the authors used 3 kinds of gray bars to present for different groups, I think they can use different colors to present different groups. In Fig. S6B, they used two kinds of red colors to present Ac-E5G2-KSYIGR-KRB and Ac-KYIGR-KRB, I think they can use black color to instead of one kind of red color.

Thanks for pointing this out! We updated the colors in Fig S6A and Fig S6B to improve clarity. (Note that is now Supplementary Figure 9.)